# To what extent does river routing matter in hydrological modeling?

Nicolás Cortés-Salazar[1], Nicolás Vásquez[1], Naoki Mizukami[2], Pablo A. Mendoza[1,3], and Ximena Vargas[1]

[1]Department of Civil Engineering, Universidad de Chile, Santiago, Chile
[2]National Center for Atmospheric Research, Boulder, CO, USA
[3]Advanced Mining Technology Center, Universidad de Chile, Santiago, Chile

*Correspondence to*: Pablo A. Mendoza (pamendoz@uchile.cl)

**Abstract.**

Spatially distributed hydrology and land surface models are typically applied in combination with river routing schemes that convert instantaneous runoff into streamflow. Nevertheless, the development of such schemes has been somehow disconnected from hydrologic model calibration research, although both seek to achieve more realistic streamflow simulations. In this paper, we seek to bridge this gap to understand the extent to which the configuration of routing schemes affects hydrologic model parameter search in water resources applications. To this end, we configure the Variable Infiltration Capacity (VIC) model, coupled with the mizuRoute routing model in the Cautín River basin (2770 km$^2$), Chile. We use the Latin Hypercube Sampling (LHS) method to generate 3500 different model parameters sets, for which basin-averaged runoff estimates are obtained directly (no routing or instantaneous runoff case), and subsequently compared against outputs from four routing schemes (Unit Hydrograph, Lagrangian Kinematic Wave, Muskingum-Cunge and Diffusive Wave) applied with five different routing time steps (1, 2, 3, 4 and 6 hours). The results show that incorporating routing schemes may alter streamflow simulations at sub-daily, daily and even monthly time scales. The maximum Kling-Gupta Efficiency (KGE) obtained for daily streamflow increases from 0.64 (instantaneous runoff) to 0.81 (for the best routing scheme), and such improvements do not depend on the routing time step. Moreover, the optimal parameter sets may differ depending on the routing scheme configuration, affecting the baseflow contribution to total runoff. Including routing models decreases streamflow values in flood frequency curves and may alter the probabilistic distribution of the medium and low flow segments of the flow duration curve considerably (compared to the case without routing). More generally, the results presented here highlight the potential impacts of river routing implementations on water resources applications that involve hydrologic models and, in particular, parameter calibration.

## 1. Introduction

Hydrology and land surface models are powerful tools to characterize the terrestrial water cycle and provide valuable information for water resources planning under future climate scenarios (Vano et al., 2012; Mendoza et al., 2016; Melsen et

al., 2018; Chegwidden et al., 2019). In applications at the catchment scale or beyond, these models are typically used in combination with river routing models that convert instantaneous runoff into realistic streamflow estimates at any locations in river networks (Oki and Sud, 1998; Olivera et al., 2000; Lucas-Picher et al., 2003). Hence, streamflow estimated by the river routing model is used for several water resources applications including flood risk assessments (Wobus et al., 2017), ecosystem

health evaluations (Qiu et al., 2021), short-term streamflow forecasting (e.g., Tang et al., 2007; Emerton et al., 2016), and reservoir operations (Salas et al., 2018; Shaad, 2018).

Over the past three decades, many river routing models have been developed and coupled with hydrology and land surface models (Shaad, 2018). The river routing models vary in terms of modeling reservoir, irrigation and other human interventions on river water (e.g., Hanasaki et al., 2006), the spatial resolution and type of discretization of the river network – grid-based

vs. vector-based (Lehner and Grill, 2013; Mizukami et al., 2016, 2021) – and, finally, the representation of flow physical processes in equations (hereafter, called routing scheme). The last category spans from a simple unit hydrograph method (Lohmann et al., 1996, 1998) to storage-based routing schemes such as Muskingum (David et al., 2011), simplifications of the Saint-Venant equations like kinematic wave (Arora and Boer, 1999; Decharme et al., 2010; Ye et al., 2013; Thober et al., 2019) or diffusive wave (Gong et al., 2009; Yamazaki et al., 2011), local inertia equations (Bates et al., 2010; Yamazaki et al.,

2013) and full dynamic wave approaches (Paiva et al., 2011).

Given the wide range of routing methods available, it is crucial to understand the benefits and limitations of each method for the specific model application (Shaad, 2018). Many studies have conducted intercomparison experiments with focus on routing schemes to evaluate their impacts on streamflow simulations. For example, Arora et al. (2001) compared a time-evolving (or variable velocity) algorithm that uses Manning's equation, against a simple storage-based routing scheme (without using

momentum equation), operating at a very different horizontal resolution. Specifically, they concluded that the variable velocity scheme can produce higher values of peak discharge. Gong et al. (2009) demonstrated the benefits of diffusive wave routing over a linear reservoir routing method to get more realistic time delays in hydrograph waves in a basin located in southern China. David et al. (2011) introduced the Routing Application for Parallel Computation of Discharge (RAPID), based on the traditional Muskingum method (McCarthy, 1938), obtaining improvements in terms of Root Mean Squared Error (RMSE) and

the Nash-Sutcliffe Efficiency (Nash and Sutcliffe, 1970) when compared to a lumped runoff scheme, which accumulate upstream instantaneous runoff without any delay. Ye et al. (2013) implemented a kinematic wave routing scheme in the Community Land Model (CLM) version 3.5, and obtained better results compared to the original grid-based River Transport Model (RTM), which uses the storage-based routing, in two basins in China.

More recently, Zhao et al. (2017) compared daily and monthly streamflow simulations produced with the CaMa-Flood

(Yamazaki et al., 2011) model – fed with daily runoff from nine Global Hydrological Models (GHMs) – against those obtained with the same hydrological models and their native routing schemes (which have simpler physics). They concluded that the choice of routing scheme may have large effects on simulated streamflow and peak values. ElSaadani et al. (2018) compared streamflow simulations obtained from VIC runoff outputs using RAPID and the Hillslope Link Model (HLM; Mantilla, 2007), which is based on power laws that relate flow velocity, channel discharge and upstream area, at many stream gauges located

in the Cedar River basin, Iowa. They noted that the choice of routing scheme has large effects on simulated hydrographs, obtaining more realistic peak times and magnitudes with the HLM model and decreasing differences in performance for larger catchments. Siqueira et al. (2018) compared a local inertia scheme against a non-hydrodynamic scheme or storage-based routing, showing that the former provided slight improvements in NSE and the Kling-Gupta efficiency (KGE; Gupta et al., 2009) over the Amazon and La Plata river basins, especially in flow timing. They highlighted that the calibration of hydrological parameters and including hydrodynamic routing are critical elements to achieve realistic streamflow simulations in South America.

Besides the complexity of the routing scheme used, the choice of routing time step may also impact streamflow calculations (Shaad, 2018). Qiu et al. (2021) characterized the effects of such decision on hydrological variables simulated with the Soil and Water Assessment Tool (Arnold et al., 1998), which uses the variable storage coefficient routing scheme, computing flow velocity with the Manning equation. The authors used six time steps ranging from 1 minute to 1 day, and assessed their impacts on performance skills including NSE and bias, finding variations in streamflow simulations that were small compared to water storages and depth.

Although many past studies have shown that the choice of routing scheme affects streamflow simulations, efforts for improving their accuracy have been made by configuring hydrologic model and routing model independently. Hydrologists still focus on parameter calibration to improve discharge simulations, excluding river routing model or neglecting the potential impacts of river routing configurations (routing scheme and time step) and parameters if included (Beck et al., 2020; Newman et al., 2021). On the other hand, routing model development and evaluation uses hydrologic model output that contains varying degree of errors, becoming especially difficult in large river basins or greater spatial domains (e.g., Mizukami et al., 2016; F. Zhao et al., 2017). Further, the key role of river routing parameters to reproduce observed streamflow characteristics has been previously recognized (Boyle et al., 2001; Butts et al., 2004; Sheikholeslami et al., 2021), highlighting the need for joint (i.e., hydrological and routing) parameter search strategies to characterize the benefits of routing configurations and potential compensatory effects in reproducing application-specific metrics.

In this paper, we seek to better understand the implications that the configuration of routing schemes may have when conducting hydrologic model calibration for water resources applications. To this end, we perform numerical experiments in the Cautín at Cajón River basin (Araucanía, Chile) using the Variable Infiltration Capacity (VIC) model (Liang et al., 1994) and the vector-based routing model mizuRoute (Mizukami et al., 2016). Specifically, we disentangle the impacts of model parameters and different routing schemes (all implemented for five time steps) by combining a large sample of VIC simulations obtained from 3500 parameter sets, and routing simulations with four different routing methods implemented in mizuRoute. Our end goal is to unravel how the choice of routing method and routing time step affect (i) streamflow simulated at different temporal resolutions, (ii) performance metrics, (iii) the selection of model parameters given a target calibration metric, (iv) simulated water balance and runoff partitioning (i.e., baseflow ratio), and (v) hydrological signatures used for decision-making, including flood frequency curves and flow duration curves (FDCs). The results and conclusions drawn here reflect the impact that innocuous modeling decisions may have for water resources management.

## 2. Study domain and data

### 2.1 The Cautín River Basin

The study domain is the Cautín at Cajón (CatC) River basin (Figure 1), a sub-catchment of the Imperial River basin, located in the Araucanía Region, Chile. The basin elevation ranges between 125 and 3104 m a.s.l., the catchment area is 2770 km$^2$, and the dominant land cover types are crop-pasture rotation (44%) and native forest (40%). Additionally, the basin is prone to rainfall-driven flood events during winter and, therefore, has been subject of studies aimed to enhance predictive capabilities 105 (e.g., Mendoza et al., 2012).

### 2.2 Hydrometeorological data

Daily precipitation, maximum and minimum temperature are obtained from the CR2MET v2.0 dataset (Boisier et al., 2018, available at https://www.cr2.cl/datos-productos-grillados/), which covers continental Chile with a horizontal resolution of 0.05° x 0.05° during the 1979-2020 period. In CR2MET, precipitation data was obtained with a statistical modeling framework 110 that uses topographic descriptors and large-scale climatic variables (water vapor and moisture fluxes) from ERA5 (Hersbach, 2016) as predictors, and observed daily precipitation from gauge stations as predictand. For maximum and minimum daily temperature, additional variables from MODIS land surface products were added as predictors. Daily precipitation and temperature time series are disaggregated into hourly time steps using the sub-daily distribution provided by ERA5-Land (Muñoz-Sabater et al., 2021). Relative humidity, wind speed and shortwave radiation are derived for the same horizontal 115 resolution grid by spatially interpolating ERA5-Land outputs. Longwave radiation was computed with the parameterization proposed by Iziomon et al. (2003), using CR2MET air temperatures disaggregated to hourly time steps using the ERA5-Land hourly distribution.

Daily streamflow data is obtained from five stations (Figure 1) maintained by the Chilean Water Directorate (DGA, available at the CR$^2$ Climate Explorer https://www.cr2.cl/datos-de-caudales/). Similarly, hourly streamflow records for the CatC basin 120 were obtained from the official DGA website (https://dga.mop.gob.cl/servicioshidrometeorologicos).

## 3. Models

### 3.1 Hydrological model

We use the VIC model (Liang et al., 1994) to simulate state variables and fluxes at a 0.05°x 0.05° horizontal resolution. VIC is a semi-distributed physically based hydrological model that solves energy and mass balance equations. Precipitation can be 125 partitioned into snowfall or rainfall, and both can be stored in the canopy. The maximum amount of water intercepted by the canopy is estimated using the Leaf Area Index (LAI; Dickinson, 1984). The soil is represented by three layers controlling the infiltration (first soil layer) and baseflow (third soil layer). For infiltration fluxes, VIC uses the Xinanjiang formulation (Zhao, 1980), assuming that the infiltration capacity varies within an area (Wood et al., 1992). Excess runoff is generated in those

areas where precipitation exceeds the amount of available soil moisture storage of the first soil layer. VIC assumes that drainage is driven by gravity, using the formulation proposed by Brooks & Corey (1964). In this regard, water enters the cell only from the atmosphere, i.e., VIC does not consider lateral fluxes among grid cells. Baseflow is generated in the third (deepest) soil layer using a formulation proposed by Franchini & Pacciani (1991). The snowpack is represented by two layers, where the top layer is used for energy balance computations (Andreadis and Lettenmaier, 2009). The reader is referred to Liang et al. (1994) for more details.

Horizontal heterogeneity is considered in each grid cell by incorporating different land cover types. Here, we use the IGBP classification for the year 2010 from the MCD12Q1 v006 land cover product (Sulla-Menashe and Friedl, 2018) to represent all land cover types spanning at least 2% of each grid cell area. Mean monthly LAI values for these land cover types are derived from the MOD15A2 product. Soil Bulk density is estimated using the mean value from the first 2 m depth of soil from the SoilGrids product (Poggio et al., 2021).

### 3.2 River routing schemes

The mizuRoute model first performs a hillslope routing using a gamma-distribution-based unit-hydrograph to delay instantaneous total runoff from the VIC model to a catchment outlet, and then route the delayed runoff for each river reach in the order defined by the river network topology. Full descriptions of hillslope routing and general routing procedures are provided in Mizukami et al. (2016). mizuRoute originally included two channel routing schemes: (1) kinematic wave tracking (KWT) routing, and (2) impulse response function (IRF) routing, which is similar to the Lohmann et al. (1996) model except that mizuRoute uses a reach-to-reach routing approach instead of the source-to-sink approach. Details of both routing schemes are also provided in Mizukami et al. (2016). Here, we implement in mizuRoute two additional routing schemes commonly used for many water resources applications: Diffusive Wave routing (DW, Appendix A) and Muskingum Cunge (MC, Appendix B). All the channel routing schemes except IRF (which uses prescribed wave celerity and diffusivity) share two parameters: Manning's n roughness coefficient and channel width (assuming rectangular channel).

### 4. Experimental setup

Figure 2 summarizes the approach used in this paper, which consists of the following steps:

   a.  Sample model (VIC and mizuRoute) parameter sets and obtain, for each one, streamflow times series with five routing schemes (including instantaneous runoff or no routing as the baseline) and five temporal resolutions (Figure 2a, see details in Section 4.1) at each river gage (Figure 1c). We save the parameter sets that maximize each performance metric – computed with a daily time step – at each stream gauge station for each combination of routing scheme and routing time step (Figure 1, Table 1). For the KWT scheme, we select *n* values of 0.01 (default option), 0.03 (i.e., the spatially constant value used by Yamazaki et al., 2011) and 0.033 (which maximizes the KGE computed with daily streamflow).

b. Examine the effect of routing model configurations (i.e., routing schemes and time steps) on simulated daily hydrographs at Cautín at Cajón (Figure 2b.1), and analyze the impact of excluding the river routing process on simulated streamflow at annual, monthly, daily and sub-daily time steps (Figure 2b.2).

c. Explore the overall impacts of routing modeling decisions on performance metrics (section 4.2) computed with different temporal resolutions (Figure 2c.1). Then, we examine the sensitivity of the best metric value (achievable from the simulations with all the sampled parameter sets) to the river routing configuration across sub-basins (Figure 2c.2).

d. Characterize the effects of routing configuration on simulated annual water balance (specifically, the mean annual runoff ratio) and baseflow contribution (computed from VIC output) to total runoff (Figure 2d.1); and the selected parameter values (Figure 2d.2).

e. Analyze the effects of routing configurations on flood frequency (see details in Section 4.3, Figure 2e.1) and daily FDCs (Figure 2e.2).

The steps (a)-(d) are performed using observed and simulated data for the period April/2008-March/2012, whereas step (e) is conducted simulations for the period April/1985-March/2020. All the steps but c.1 (Figure 2) use VIC and mizuRoute parameter sets that maximize performance metrics (listed in Section 4.1) computed with simulated and observed daily flows. The following sub-sections provide complete descriptions on the parameter sampling strategy, streamflow simulations, performance metrics and the computation of flood frequency and flow duration curves.

### 4.1 Parameter sampling and streamflow simulations

Since we aim to examine the impacts of different routing schemes on streamflow performance metrics across the parameter space, rather than seeking an optimal parameter set, we use the Latin Hypercube Sampling (LHS) method, which is a common strategy to sample the parameter space and identify behavioral sets for specific target metrics (Andréassian et al., 2014; Broderick et al., 2016; Melsen et al., 2016, 2019; Guse et al., 2017; Khatami et al., 2019). Here, we sample 3500 model parameter sets (Figure 2a) considering the 13 VIC parameters identified by Sepúlveda et al. (2022) as the most sensitive, and routing model parameters: one (the Manning roughness coefficient) for the KWT, MC and DW methods, and two for the IRF method (Table 2). For each parameter set, we run VIC and mizuRoute at hourly time steps for the period April/2006 - March/2012. To generate streamflow simulations at 2-hour, 3-hour, 4-hour, and 6-hour time steps, we aggregate hourly VIC runoff and run the mizuRoute model with all routing schemes. For example, streamflow time series at a 3-hour resolution are obtained from mizuRoute simulations using 3-hour VIC runoff time series, which are computed by temporally aggregating 1-h VIC outputs at each grid cell. It should be noted that this step requires assuming the absence of non-linear processes in time within the hydrological model. The resulting VIC runoff time series are also used to compute streamflow by spatially averaging total runoff within each (sub-)basin without using a routing scheme (hereafter referred to as instantaneous runoff, Inst, or no routing), which is a common approach used in hydrological modeling applications (e.g., Mendoza et al., 2016; Beck et al.,

2020). As a result, we obtain streamflow times series at each river reach (Figure 1c) for five routing methods (four routing schemes and Inst as the baseline) and five temporal resolutions (1 h, 2 h, 3 h, 4 h and 6 h, Figure 2a).

### 4.2 Streamflow performance metrics

For each model run, we evaluate the performance of streamflow simulations from VIC-mizuRoute using four metrics. The first one is the Kling Gupta efficiency (KGE; Gupta et al., 2009; Kling et al., 2012), which quantifies performance in terms of variability, volume and timing:

$$KGE(Q) = 1 - \sqrt{(1-\alpha)^2 + (1-\beta)^2 + (1-r)^2} \qquad \alpha = \frac{\sigma_s}{\sigma_o} \qquad \beta = \frac{\mu_s}{\mu_o} \qquad (1)$$

where $\sigma$ is the standard deviation for simulated and observed values, $\mu$ is the mean streamflow over the $n$ times steps, and $r$ is

200 the Pearson correlation coefficient between simulated and observed streamflow. The second metric is the Nash-Sutcliffe efficiency (NSE; Nash & Sutcliffe, 1970), which is computed using observed (o) and simulated (s) streamflow ($Q$):

$$NSE(Q) = 1 - \frac{\sum_{t=1}^{n}(Q_o^t - Q_s^t)^2}{\sum_{t=1}^{n}(Q_o^t - \bar{Q}_o)^2} \qquad (2)$$

where $Q_o^t$ is the observed streamflow for time step $t$, $Q_s^t$ is the simulated streamflow for time step $t$ and $\bar{Q}_o$ is the mean observed streamflow over the $n$ time steps considered. The third metric is the NSE computed for the logarithms of the

205 streamflow (NSE-log) to test the model's capability to simulate low flows (Krause et al., 2005). Although all these metrics range between $-\infty$ and 1, where 1 represents a perfect simulation, their values are not comparable because they differ in terms of target streamflow characteristics, and the incorporation or lack of a benchmark. For example, NSE is formulated based on a reference model simulation (i.e., mean climatology), while KGE does not have one, and NSE = 0 is equivalent to KGE = - 0.41 (Knoben et al., 2019). Importantly, the above metrics are relevant not only for the hydrologic modeling community –

210 especially for parameter calibration and evaluation (Fowler et al., 2018; Knoben et al., 2019; Clark et al., 2021) –, but also for the river routing community. In fact, several examples of river routing scheme assessments can be found using the KGE (Pereira et al., 2017; Hoch et al., 2019; Qiao et al., 2019; Thober et al., 2019; Munier and Decharme, 2022), NSE (Yamazaki et al., 2011; Ye et al., 2013; Zhao et al., 2017; ElSaadani et al., 2018; Nguyen-Quang et al., 2018; Fleischmann et al., 2019, 2020) and even the NSE-log (Paiva et al., 2013b).

Finally, we use the percent bias in the high-segment volume of the FDC (Yilmaz et al., 2008):

$$\%BiasFHV = \frac{\sum_{h=1}^{H}(Q_s^h - Q_o^h)}{\sum_{h=1}^{H} Q_o^h} \times 100 \qquad (3)$$

where h =1,2,...,H are the flow indices in the flow array with probability of exceedance less than 0.02. FHV is a measure of the basin response to high precipitation events.

The four performance metrics are calculated for the period April/2008 – March/2012 (after a two-year warm up), using all the

220 combinations of parameter sets (3500), routing schemes (including the case without routing) and routing time steps (1 h, 2 h, 3 h, 4 h and 6 h). Additionally, for each routing time step, the performance metrics are computed for different aggregated temporal resolutions when possible for step (c.1). For example, to estimate metrics at an hourly time step, routing can only be

run at a 1-hour time step. Metrics computed at 3-hourly time steps use temporally averaged streamflow from a 1-hour and 3-hour mizuRoute simulations. Metrics computed at 6-hourly time steps can be computed from temporally averaged 1-hour, 3-hour and 6-hours mizuRoute simulations, and so on. The observed streamflow for a given time step is estimated from hourly streamflow records.

### 4.3 Flood frequency and flow duration curves

Because high flows are relevant for engineering applications, in particular, infrastructure design, we analyze the implications of routing configurations for the calculation of flood frequency curves (Figure 2e.1). To this end, we run VIC at hourly time steps from April/1981 to March/2020 using the parameters associated to the highest KGE, NSE and %BiasFHV values (all computed with daily flows) for each routing configuration. Then, hourly VIC total runoff is aggregated and routed at different time steps (i.e., 2 h, 3 h, 4 h and 6 h), and annual maximum daily flows are obtained for the period April/1985 – March/2020 (i.e., the period April/1981 – March/1985 is dropped). Hence, for each routing time step we obtain five annual time series with n = 35 values (obtained from the baseline and the four routing schemes) that are used to compute maximum daily flows at return periods of 20, 50, 100 and 200 years. We use the Log-Normal parametric distribution – which provides the best results for the Kolmogorov–Smirnov test – for the observed time series of maximum daily flows.

Finally, we characterize the impacts of routing configurations on daily FDCs (Figure 2e.2), which are widely used in water resources applications. Empirical FDCs are constructed from daily time series of streamflow for the period April/1985 – March/2020, computing exceedance probabilities with the Weibull plotting position formula.

## 5. Results

### 5.1 Illustration of routing effects

Figure 3 illustrates the sensitivity of daily streamflow simulations to different routing modeling decisions, including the routing time step on IRF scheme (Figure 3a), the choice of routing scheme (Figure 3b), and the Manning's roughness coefficient on KWT results (Figure 3c). In each panel, simulations are displayed for the period May 10 – June 16 2008 using the parameter set (obtained from LHS) associated with the maximum daily KGE for each combination of routing scheme, routing time step and Manning's coefficient distribution. In all cases, sub-daily routing simulations are aggregated to a 24-hour time step. It can be noted that the choice of routing scheme and Manning's coefficient values have a larger effect on the shape of the flood wave. Additionally, increasing routing time steps for IRF accelerates the timing of peak discharge in one day, though decreasing its value to 776 m$^3$/s for $\Delta t$ = 6 h (compared to 785 m$^3$/s obtained with $\Delta t$ = 1 h). The choice of routing scheme affects the shape of storm hydrographs, especially high flows. Finally, a delay in peak flow simulations is obtained for larger Manning's roughness coefficients.

Figure 4 compares streamflow obtained from mizuRoute (y-axis) against instantaneous runoff (x-axis) for several temporal resolutions and different routing schemes. In this case, the parameter set used to run the models is the one that maximizes the

KGE among the 3500 parameter sets from the LHS. The results for hourly time steps show that the lack of routing yields much larger values (> 1300 m$^3$/s in some cases) compared to routed streamflow. These differences are gradually reduced when the routing time step increases to $\Delta t = 3$ h and 6 h, although differences can be larger than 1200 m$^3$/s. The impact of excluding routing lessens as the time step increases, yet it can be important even for streamflow simulations at a $\Delta t = 24$ h time step. At monthly time steps, the differences between routed and instantaneous runoff reduce considerably, though these still can be as large as 27 m$^3$/s (i.e., a 10% difference using routed runoff as the reference). Further, the differences become negligible at the annual resolution. Finally, given a specific time step, the magnitudes of differences are very similar across routing schemes, although slight differences in r$^2$ suggest that IRF and KWT affect VIC outputs more.

## 5.2 Effects on performance metrics

The KGE, NSE, NSE-log and %BiasFHV values obtained with the 1% best (i.e., 35) parameter sets for each combination of routing scheme, routing time step and metric time step are displayed in Figure 5. To compare performance measures from different configurations, simulations were aggregated to the metric time resolution (columns). For the sake of brevity, we only show 1 h and 24 h metric time steps here (the full results can be found in Figure S1). Overall, the results show a clear difference between including routing and Inst, which becomes more evident for performance metrics computed at smaller temporal resolutions. Moreover, none of the 1% best parameter sets for KGE and NSE with instantaneous runoff could produce better performance than including routing. On the other hand, the choice of routing time step is comparatively less influential for a given metric time step. The maximum KGE spans 0.69-0.73 for instantaneous runoff, increasing to 0.8 or more when routing is included. Similar improvements are observed for NSE, with increments that can be larger than 0.3 NSE values (e.g., 1 h). Compared to the former metric, routing yields smaller benefits for NSE-log and less noticeable differences among routing configurations, mainly due to the minor influence of high flow values on the metric. In all cases, a larger spread in high-flow biases is obtained with instantaneous runoff (compared to routing schemes), indicating that many VIC parameter sets do not compensate for the lack of river routing to obtain accurate high flow simulation. Finally, the results show that the impact of representing river routing on %BiasFHV is more relevant when this metric is computed at hourly resolution, approaching to zero as the metric time step increases.

Figure 6 compares the best KGE, NSE, NSE-log and %BiasFHV values (computed from daily flows) achievable from the large sample of model parameter sets in each basin (represented by the basin area in the x-axis), given a specific combination of routing scheme and routing time step. For completeness, the KGE components ($\alpha$, $\beta$ and $r$) are also displayed. For KGE, NSE and NSE-log, the maximum values increase at all streamflow gages when the routing process is included, regardless of the routing configuration. Very similar maximum KGE values are obtained with the four schemes implemented in mizuRoute, and the differences among these schemes are generally lower than 0.05 for all time steps and basins. The improvements in KGE through the inclusion of routing are explained by the enhancement of temporal correlation ($r$) and variability error ($\alpha$). In particular, routing (especially Muskingum-Cunge and Diffusive Wave schemes) helps to improve $r$ values of simulated

instantaneous runoff by changing the timing of high peak flows. Even more, Figure 6 shows that, in our study domain, the correlation between streamflow simulations and observations increases with basin area when routing is incorporated.

Figure 6 also shows considerable improvements in NSE across all catchments when routing is applied, especially in CatRR and CatC (i.e., the two largest river basins). Notably, differences between routing and Inst options are also obtained for NSE-log in all stream gages, demonstrating the benefits of routing beyond the simulation of high flows. %BiasFHV is very close to zero for the smallest catchment, increasing with catchment size when no routing is performed; nevertheless, in every basin, at least one routing scheme can reduce high flow biases to nearly zero.

### 5.3 Impact on simulated fluxes and model parameters

For the remainder of this paper, we illustrate the impacts of routing time step by showing results for 1 h, 3 h and 6 h (the full results are available in the Supporting Information). Figure 7 illustrates the impacts of the choice of routing scheme on the mean annual runoff ratio (partitioning of precipitation into runoff and evapotranspiration: x-axis) and the ratio between mean annual baseflow and mean annual total runoff (y-axis) for each routing time step (columns) and performance metrics (all computed with daily discharge and displayed in different rows). The red symbols represent the results based on the best value for the corresponding performance metrics. To account for equifinality effects, we also include the results with the parameter sets that produce the 0.1% best simulations for respective performance metrics (i.e., four parameter sets). Precipitation partitioning (Q/P) is relatively unchanged whether routing is used or not, regardless of the routing schemes for any performance metric except %BiasFHV, for which less clear patterns in runoff ratio and flow components are obtained. Conversely, the greater impacts are seen in the baseflow ratio (bf/Q). For KGE and NSE, a clear separation in runoff components is obtained between instantaneous runoff and routing schemes (for any routing time step), with a much higher baseflow contribution to total runoff for Inst. When the VIC model parameters are selected based on the NSE-log metric, the differences among the routing configuration options are generally smaller compared to KGE and NSE, though the best performing parameter set produces less baseflow.

To examine the effect of river routing on the selection of the model parameters, we show, for three routing time steps and all routing schemes, the best values for KGE, NSE, NSE-log and %BiasFHV (with the four metrics computed with daily flows) among the 3500 parameter sets from LHS (Figure 8). The parameters values are normalized by the difference between the maximum and minimum values obtained from LHS to facilitate comparisons. Hence, a normalized value of zero indicates the lower boundary of the parameter, while a value of 1 indicates the upper limit. The results indicate that the same best parameter set is obtained for NSE-log regardless of the selected routing scheme (when included) or the routing time step, which explains why all the routing schemes with the best performing parameter set (red) produce the same bf/Q and Q/P (Figure 7). Additionally, the absence of routing when maximizing NSE-log not only affects soil parameter values (compared to results with river routing), but also reduces the snow albedo decay parameter ($\alpha_{thaw}$).

For KGE, the KWT, MC and DW schemes yield the same VIC parameter sets, which are different from that obtained with instantaneous runoff. Conversely, all the routing schemes yield different VIC parameter sets for NSE, except for MC and DW.

For KGE and NSE, the choice of routing time step does not affect the best parameter set given a routing scheme, excepting the combination KGE-IRF. It should be noted that, for both NSE and KGE, excluding routing produces higher $W_s$ (fraction of maximum soil moisture where non-linear baseflow occurs), higher $d_{max}$ (maximum total soil thickness) and lower $b$ (infiltration parameter), regardless of the routing time step, which may contribute to the larger baseflow fraction seen in Figure 7. Finally, different routing configurations (routing methods and routing time step) result in unique best parameter sets if %BiasFHV is used as the performance metric, in contrast to KGE, NSE and NSE-log.

### 5.4 Implications for flood frequency and flow duration curves

Figure 9 shows the flood frequency curves from the annual time series of maximum daily flows, using model parameters obtained with KGE, NSE and %BiasFHV as target metrics. Note that the curve for daily instantaneous runoff is the same for each metric, regardless of the time step. As expected, differences in flood estimates between instantaneous runoff and any other routing scheme are considerable, surpassing 400 m³/s in some cases (see results for T = 200 with KGE and %Bias FHV). Additionally, the dispersion among routing schemes increases with larger Δt for KGE and, in particular, for %BiasFHV, which can be explained by differences obtained for model parameter values (Figure 8). Even if the same parameter sets are obtained for a target metric and a suite of routing schemes, regardless of the choice of Δt (e.g., KWT, MC, and DW for KGE, see Figure 8), variations in routing time step affect the time series of daily flows and, therefore, flood quantiles obtained with a specific routing scheme, as well as inter-method differences for a given return period. For example, when the target metric is KGE, Q(T=100 years) values obtained with KWT and DW for Δt = 1 h are 1202 m³/s and 1189 m³/s, respectively, while Δt = 6 h yields 1196 m³/s and 1171 m³/s using the same schemes. Interestingly, differences in frequency curves arising from the choice of routing scheme with NSE decrease with larger Δt, although the best parameter set does not depend on the routing scheme (see Figure 8).

Figure 10 shows daily FDCs obtained with different routing schemes, routing time steps (columns) and model parameters that maximize KGE, NSE, or %BiasFHV (rows). It can be noted that, for KGE, the disagreement arising from the choice of routing scheme is generally small for medium and low flows, as opposed to the disagreement obtained with NSE and %BiasFHV. For KGE and NSE, no appreciable differences in FDCs are observed among routing time steps, whereas the opposite is observed for %BiasFHV. For example, FDCs with very different mid-segment slopes (which is a signature for flashiness of runoff) and low flow volumes (i.e., segment for $P_{exc} > 70\%$) are obtained for MC with Δt = 3 and Δt = 4 h (see Figure S4). This can be explained by the fact that the choice of routing time step does not impact the parameter sets obtained with NSE and KGE (as it happens with %BiasFHV).

## 6. Discussion

### 6.1 Implications for hydrological modeling

In this paper, we use the LHS approach to evaluate the impact of routing on streamflow performance metrics across the
parameter space. Our results suggest that, regardless of the routing scheme, including the river routing process improves the overall streamflow performance (Figures 5 and 6). In other words, the lack of river routing in the modeling chain may not be fully compensated through the calibration of hydrologic model parameters. Nevertheless, such conclusion may depend on the hydrological regime of the catchment and the distributed spatial configuration of the river routing implementation. The Cautín River basin has a rainfall-dominated runoff regime, with high flow peaks associated with heavy rainfall events during the
winter season, and a slight influence of snowmelt during the spring season. The catchment response time and peak discharge depend on the runoff routing process in the river network; hence, its explicit inclusion in hydrological modeling may yield better results, especially for performance metrics influenced by high flows (Clark et al., 2021).

The results presented here show that the implementation of river routing is also relevant for medium and low flows. For example, including river routing provided higher values for NSE-log (Figures 5 and 6) – improving the simulation of low
discharges – and modified the shape of the mid and low flow segments in the FDC (Figure 10), which are characteristic signatures of 'flashiness' in runoff response and long term baseflow, respectively (Yilmaz et al., 2008). The effects of river routing are also reflected in the partitioning of total runoff between baseflow and surface runoff. In fact, the results presented here show that the parameter search process compensates for the lack of routing by modifying other fluxes and state variables (Khatami et al., 2019) to increase streamflow-oriented performance metrics. For example, when the target metrics are KGE,
NSE or NSE-log, excluding routing (represented by squares) forces VIC to compensate the absence of this process by delaying the runoff response with a larger contribution of baseflow to total runoff, compared to any routing schemes. In such cases, the contribution of baseflow to total runoff increases >30% when river routing is excluded, which is achieved by modifying soil parameters –including $W_s$, one of the most sensitive for baseflow in this type of hydroclimate (Sepúlveda et al., 2022) – to delay the streamflow response. This result suggests that including routing processes may impact the outcomes from drought-
oriented studies, since baseflow is the primary flux sustaining streamflow during water scarcity periods (Karki et al., 2021). Conversely, we found smaller variations in the partitioning of precipitation between evapotranspiration and runoff in the absence of river routing (Figure 7), especially for KGE, NSE and NSE-log. Hence, when models are configured to maximize these metrics to conduct hydroclimatic or water balance analyses at the annual time scale, the incorporation of routing processes is relatively less important.

The impacts of routing scheme choice exhibit less clear patterns if the model chain is calibrated with integrated time series metrics such as KGE, though differences remain in the performance metrics (Figure 5), high flow analyses (Figure 9) and FDCs (Figure 10). Here, we obtained very similar results with MC and DW, including runoff partitioning, best parameter sets (including Manning's roughness coefficient *n*) and flood frequency curves if the target metrics are KGE and NSE. Both schemes use the same routing parameters and essentially simulate wave diffusion. MC mimics physical diffusive phenomena

via numerical diffusion in the explicit numerical solution, while the DW routing explicitly incorporates the diffusion process in the diffusion equation. Despite very good results were achieved with DW, limited impacts are expected for the Cautín River basin, because the slopes of river reaches therein range from 0.0004 to 0.274 m/m. In fact, the largest benefits of DW are expected for flatter river systems (slope < 0.001 m/m; e.g., Kazezyılmaz-Alhan et al., 2007), where flood wave diffusion processes can dominate. It can be argued that a more physically realistic routing scheme will better simulate the hydrograph.

However, IRF routing, which uses the simplest algorithm among the schemes used in this study, may reproduce the results from the other routing schemes after the calibration.

    Although the routing time step does not yield important effects on performance metrics in our experimental setup, it can affect the choice of VIC parameters (e.g., see results for KGE and %BiasFHV, Figure 8), which is in line with previous hydrologic modeling research. For example, Kavetski et al. (2011) found that temporal data resolution may alter parameter values in

conceptual hydrological models. More recently, Melsen et al. (2016) found that the parameter values may greatly vary if calibration metrics are computed at hourly, daily or monthly time steps. Accordingly, variations in the VIC model time step – which is fixed to $\Delta t = 1$ h here – may also alter the selection of parameters and performance measures (see section 6.2).

### 6.2 Limitations and future work

    Here, we only focused on the choice of routing scheme and routing time step, though there are many other decisions that could

be explored in the implementation of river routing models. For example, we did not examine the effects of surface storage elements like reservoirs, wetlands, and flood plains on river flow dynamics. Additionally, we did not estimate the spatial variability of the Manning's roughness coefficient ($n$) across the Cautín River basin. Due to data restrictions, many past studies used spatially constant values of $n$ (Arora and Boer, 1999; Lucas-Picher et al., 2003; Yamazaki et al., 2011; Siqueira et al., 2018), or have adopted indirect approaches. For instance, Decharme et al. (2010) estimated $n$ as a linear function of the river

width W; Miguez-Macho & Fan (2012) used satellite land cover to assign the Manning's roughness coefficient and Verzano et al. (2012) estimated $n$ variability in space based on topography, the location of urban population, and river sinuosity. These or other techniques could be applied in combination with field data to estimate spatial $n$ fields, that can be subsequently calibrated through spatial regularization strategies (e.g., Mendoza et al., 2012).

    An important assumption of our experiment is the lack of non-linear processes in time within the hydrological model, in order

to aggregate hourly runoff to coarser time steps. Such decision was required to isolate the impact of hydrologic model configuration from river routing decisions, and achieve a clean experimental design, though we recognize that the choice of hydrological model time step may also alter performance metrics (e.g., Bruneau et al., 1995; Wang et al., 2009).

    In this study, we did not include the full dynamic wave scheme, which might yield improvements compared to the routing schemes tested here, especially very large flood events at downstream of the bases or flatter part of basins. Paiva et al. (2013b)

validated a full hydrodynamic model in stream gauges within the Amazon River basin, obtaining that discharge and water levels were simulated accurately, outperforming the Muskingum-Cunge approach. The same model was evaluated against satellite observations, showing good performance in terms of water levels and inundation extents (Paiva et al., 2013a). Hence,

future assessments of routing schemes may include more detailed comparisons against remotely sensed data, including such additional hydraulic variables over ungauged catchments with different hydrological regimes (e.g., snowmelt-driven, mixed regimes) and physiographic characteristics (e.g., contributing area, average slope, land cover types). Further, it would be interesting to examine the interplay between structural uncertainty (e.g., channel geometries, rive network or drainage density, and floodplain) and parametric uncertainty in river routing models, a topic that has been widely explored in the hydrologic modeling literature (e.g., Ajami et al., 2007; Günther et al., 2019; Newman et al., 2021).

In this work, we contrast the performance of different model configurations using metrics that describe specific aspects of the system's response. However, the interpretation of differences in metric values is not straightforward (Clark et al., 2021). For example, an improvement of 0.2 in KGE may be explained by a better simulation of streamflow timing and variability, at the cost of larger volume bias (e.g., see results for Cautín at Rariruca, 1305 km$^2$, Figure 6), which stresses the need to understand the information content in the metrics used for model diagnostics. Finally, we only considered a single-objective (e.g., NSE, KGE) parameter search based on a Monte Carlo sampling scheme. Future studies could characterize the impacts of river routing schemes exploiting single-objective optimization algorithms (e.g., Duan et al., 1992; Tolson and Shoemaker, 2007), or address multi-objective problems using Pareto principles (Yapo et al., 1998; Pokhrel et al., 2012; Shafii and Tolson, 2015). Although different behavioral parameter sets and, therefore, different internal fluxes could be obtained, we hypothesize that similar conclusions could be drawn regarding the benefits of river routing representation to achieve realistic streamflow simulations. Nevertheless, further research is needed to understand implications for catchments with different hydroclimatic regimes and physiographic characteristics (e.g., Murillo et al., 2022; Muñoz-Castro et al., 2023).

## 7. Conclusions

Despite the general consensus in the hydrology and Earth system modeling communities about the relevance of river routing schemes for realistic streamflow simulations, there is little knowledge of the extent to which this process is relevant. Additionally, hydrologic model calibration research has been done neglecting the impacts of river routing model configurations, and routing model development has been conducted ignoring the effects of hydrologic model parameters. In this paper, we try to reduce these gaps by performing modeling experiments at the Cautín River basin (Chile), coupling the VIC model with four different routing schemes implemented in the mizuRoute model to produce streamflow simulations at various time steps with an ensemble of 3500 parameter sets.

Our main conclusions are as follows:

1. Runoff routing alters streamflow simulations considerably at sub-daily and daily time steps, with slight (negligible) impacts at the monthly (annual) time step.
2. Including a river routing model may provide better hydrologic model calibration results compared to the case without routing.
3. The timing of streamflow simulations may improve for larger contributing areas if runoff routing is performed.

4.  For popular performance metrics (i.e., KGE, NSE and NSE-log), including routing processes may yield different parameter sets compared to the case without routing, with notable impacts on the baseflow contribution to total runoff. Additionally, different routing schemes may yield different hydrologic parameter sets.

5.  Including routing models decreases annual maximum daily flows values in frequency curves and, depending on the target streamflow metric, the disagreement in flood quantile estimates among schemes may increase for larger routing

time steps.

6.  When the calibration metric is NSE($Q_{24h}$) or %BiasFHV, including routing models may affect the probabilistic distribution of medium and low daily flows considerably.

## Appendix A. Diffusive wave routing

The flood wave propagation through a river channel is described with the 1-dimensional Saint-Venant equations:

$$\frac{\partial Q}{\partial x} + \frac{\partial A}{\partial t} = 0 \tag{Eq. A1}$$

$$\frac{\partial Q}{\partial t} + \frac{\partial}{\partial x}\left(\frac{Q^2}{A}\right) + gA\frac{\partial Z}{\partial x} - gA(S_o - S_f) = 0 \tag{Eq. A2}$$

where $Q$ is discharge [$L^3T^{-1}$] at time step $t$ [T] and location $x$ [L] in a river reach, $A$ is cross-sectional flow area [$L^2$], $Z$ is flow depth [L], $S_0$ is channel slope [-], $S_f$ is friction slope [-], and $g$ is gravitational constant [$LT^{-2}$]. The continuity equation (Eq .A1) assumes that no lateral flow is added to a channel segment. A friction slope is expressed using channel conveyance $K_c$:

$$S_f = \frac{Q|Q|}{K_c} \tag{Eq. A3}$$

In large domain river routing, one-dimensional full Saint-Venant equations, or fully dynamic wave equations, are typically simplified by neglecting some force terms in the momentum equation (Eq. A2). The kinematic wave approximation is obtained by neglecting acceleration and pressure gradient terms, assuming that river bed slope and energy slope are equal. This assumption is the basis of the kinematic wave tracking algorithm (Mizukami et al., 2016). If a rectangular channel with a

channel width $w$ is used, the diffusive wave equation can be obtained by neglecting acceleration terms (1st and 2nd terms in Eq. A2) and combining Eqs. A1 and A2 (Sturm, 2021):

$$\frac{\partial Q}{\partial t} = D\frac{\partial^2 Q}{\partial x^2} - C\frac{\partial Q}{\partial x} \tag{Eq. A4}$$

Where:

$$C = \frac{1}{K_c}\frac{dK_c}{dA} = \frac{dQ}{dA}$$

$$D = \frac{K_c^2}{2qw} = \frac{Q}{2wS_o}$$

where $K_c$ is conveyance, and parameters $C$ and $D$ are wave celerity [$LT^{-1}$] and diffusivity [$L^2T^{-1}$], respectively.

To solve the diffusive wave equation for discharge $Q$, Eq. A4 is discretized using weighted averaged finite difference approximations across two time steps in space (i.e., $2^{nd}$-order central difference in the $1^{st}$ term in A4, and $1^{st}$ order central difference for $2^{nd}$ term in A4). The resulting discretized diffusive wave equation is:

$$(\alpha C_a - 2\beta C_d) \cdot Q_{j+1}^{t+1} + (2 + 4\beta C_d) \cdot Q_j^{t+1} - (\alpha C_a + 2\beta C_d) \cdot Q_{j-1}^{t+1}$$

$$= -[(1-\alpha)C_d - 2(1-\beta)C_d] \cdot Q_{j+1}^{t} + [2 - 4(1-\beta)C_d] \cdot Q_j^{t} + [(1-\alpha)C_a + 2(1-\beta)C_d] \cdot Q_{j-1}^{t}$$

$$C_a = \frac{C \cdot \Delta t}{\Delta x} \qquad C_d = \frac{D \cdot \Delta t}{(\Delta x)^2} \qquad \qquad \text{(Eq. A5)}$$

Where $\alpha$ is weight factor for the $1^{st}$ order space difference approximation of the second term in Eq. A4, and $\beta$ is a weight factor for the $2^{nd}$ order space difference approximation of the first term in Eq. A4. If both weights are set to 1, the finite difference becomes a fully implicit scheme, while setting both weights to zero results in a fully explicit scheme.

If internal nodes are defined within each reach (here we used 5), Eq. A5 becomes a system of linear equations that can be expressed in tridiagonal matrix form and solved with the Thomas' algorithm. In this paper, we use a fully implicit finite difference approximation (i.e., $\alpha = \beta = 1$). The solution of the implicit method requires downstream and upstream boundary conditions, being the latter inflow from upstream reaches. We use the Neumann boundary condition, which specifies the gradient of discharge between the current and downstream reaches. Note that in diffusive wave routing, celerity ($C$) and diffusivity ($D$) are updated at every time step based on the discharges ($Q$) and flow area ($A$), as opposed to IRF routing in which celerity and diffusivity are provided as model parameters.

**Appendix B. Muskingum-Cunge**

In the Muskingum-Cunge (MC) routing approach, the desired streamflow value is computed as the weighted ($C_1$, $C_2$, and $C_3$) average of known discharge values at upstream and downstream positions, at current and previous time steps:

$$Q_{j+1}^{t+1} = C_1 \cdot Q_j^{t} + C_2 \cdot Q_{j+1}^{t} + C_3 \cdot Q_j^{t+1} \qquad \qquad \text{(Eq. A6)}$$

$$C_1 = \frac{2KX + \Delta t}{2K(1-X) + \Delta x} \qquad C_2 = \frac{2K(1-X) - \Delta t}{2K(1-X) + \Delta x} \qquad C_3 = \frac{-2KX + \Delta t}{2K(1-X) + \Delta x}$$

The parameters $K$ and $X$ are defined as;

$$K = \frac{\Delta x}{C} \qquad X = 0.5 - \frac{Q}{2S_o C \Delta x}$$

Here, both parameters are computed with discharge $Q$ updated at every time step based on the average of inflow at the current time step and inflow and outflow at the previous time step. Note that celerity is also a function of discharge. Since Muskingum-Cunge is an explicit method, the routing time step can affect the numerical stability of the solution. To stabilize the solution, sub-routing time step is determined at every simulation step so that Courant condition (C*dT/dx where C is wave celerity [L/T], dT is routing time step [T] and dx is channel length [L]) is less than unity.

## Code availability

The codes and data used in this study are publicly available for download at Zenodo (Cortés-Salazar et al., 2023 https://doi.org/10.5281/zenodo.7838673).

## Author contributions

All the authors were involved in the conceptualization of this study. NC, NV, NM and PM designed the methodology and analysis framework and drafted the paper. NV configured the VIC model. NM developed all the routing schemes in mizuRoute. NC configured mizuRoute, conducted simulations, analyzed the results and created the figures. XV provided insights into the analysis results. All the authors discussed the results and contributed to writing, reviewing and editing the manuscript.

## Competing interests

The authors declare that they have no conflict of interest.

## Acknowledgments

We thank the editor (Markus Hrachowitz) and two anonymous reviewers for their constructive comments, which helped to improve this paper considerably.

## Financial support

Nicolás Cortés-Salazar, Nicolás Vásquez and Pablo A. Mendoza received support from Fondecyt Project 11200142; Pablo A. Mendoza was also supported by CONICYT/PIA Project AFB220002. The National Center for Atmospheric Research is a major facility sponsored by the National Science Foundation under cooperative agreement no. 1852977.

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

**Table 1. Stream Gauge Stations in the Cautín at Cajón River basin. Annual streamflow at each station was obtained from daily records for the period April 1985-March 2020.**

| | Station Name | Abbreviation | Latitude (°S) | Longitude (°W) | Area (km$^2$) | Elevation (m a.s.l.) | Mean Annual Flow (m$^3$/s) |
|---|---|---|---|---|---|---|---|
| 1 | Collín at Codahue | CatCd | 38.58 | 72.19 | 259 | 250 | 12 |
| 2 | Muco at Muco Bridge | MatPM | 38.61 | 72.39 | 651 | 250 | 24 |
| 3 | Cautín at Rariruca | CatRR | 38.43 | 72.01 | 1305 | 425 | 86 |
| 4 | Cautín at Cajón | CatC | 38.69 | 72.50 | 2770 | 130 | 130 |

**Table 2. Model parameters sampled in this study.**

| Parameter | Units | Lower value | Upper value | Description |
|---|---|---|---|---|
| Infilt | - | 0.01 | 0.99 | Variable infiltration curve parameter |
| $D_s$ | - | 0.1 | 0.9 | Fraction of $D_{s\,max}$ where non-linear baseflow occurs |
| $D_{s\,max}$ | mm/d | 0.1 | 300 | Maximum velocity of baseflow |
| $W_s$ | - | 0.1 | 0.9 | Fraction of maximum soil moisture where non-linear baseflow occurs |
| expt | - | 3.1 | 10 | Exponent of Campbell's equation for hydraulic conductivity |
| $d_{max}$ $d_1$ $d_2$ $d_3$ | m | 0.5 $0.05\,d_{max}$ $0.21\,d_{max}$ $0.74\,d_{max}$ | 5 $0.2\,d_{max}$ $0.7\,d_{max}$ $0.1\,d_{max}$ | Depth of soil layers 1, 2 and 3 |
| $K_{sat}$ | mm/d | 1 | 1000 | Saturated hydraulic conductivity |
| $T_{max,snow}$ | (°C) | -10 | 10 | Maximum temperature for snowfall |
| $\alpha_{thaw}$ | - | 0.75 | 0.90 | Decay of albedo |
| $\alpha_{new}$ | - | 0.85 | 0.95 | Maximum albedo for fresh snow |
| $n$ | $s/m^{1/3}$ | 0.024 | 0.075 | Roughness coefficient of Manning (Barnes, 1967) |
| $C$ | $m/s$ | 0.25 | 10 | Advection coefficient (Allen et al., 2018) |
| $D$ | $m^2/s$ | 200 | 4000 | Diffusion coefficient (Melsen et al., 2016) |

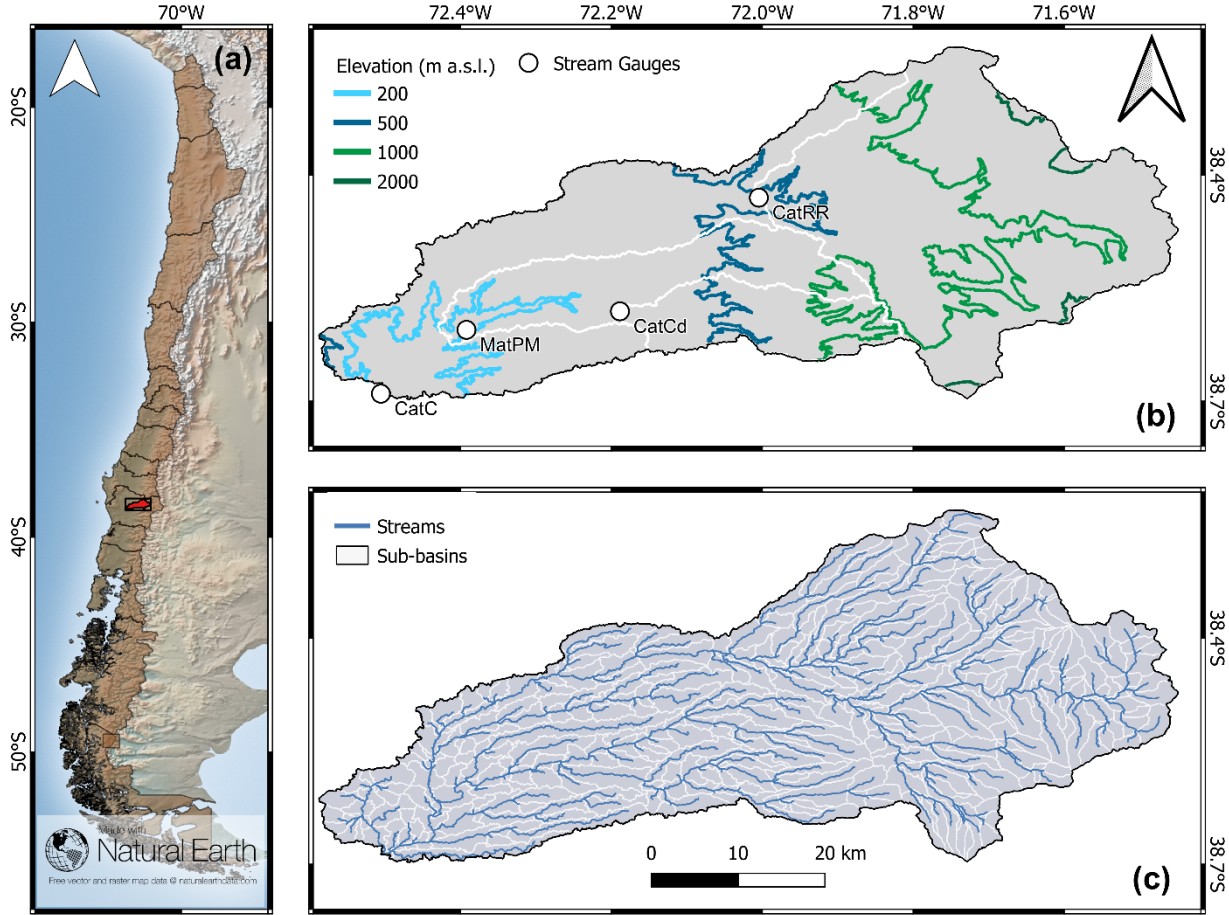

**Figure 1. (a) Location of the Cautín at Cajón River basin in Chile (CatC, 2770 km²). (b) Location of outlet and inner stream gauge stations (white circles) and contributing drainage areas (white lines). The inner stations are Muco at Muco bridge (MatPM, 651 km²), Collín at Codahue (CatCD, 259 km²) and Cautín at Rariruca (CatRR, 1305 km²). (c) Digital river network and sub-basin boundaries used in mizuRoute.**

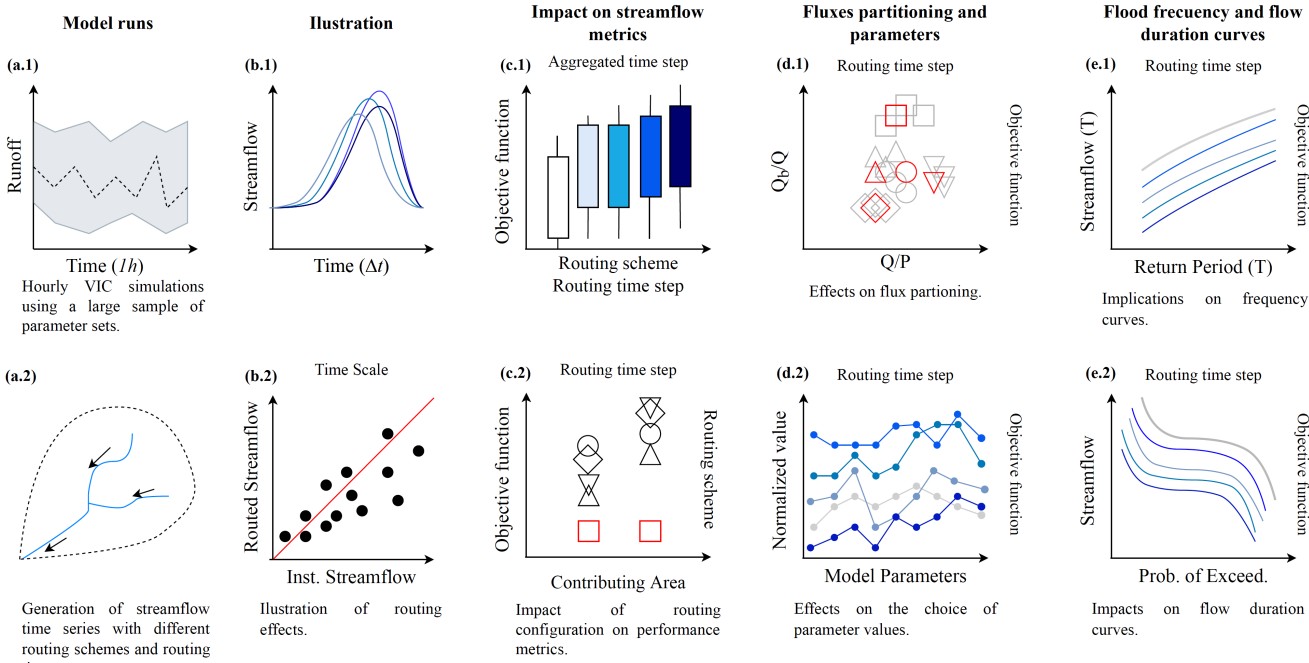

**Model runs**

**(a.1)**
Hourly VIC simulations using a large sample of parameter sets.

**(a.2)**
Generation of streamflow time series with different routing schemes and routing time steps.

**Ilustration**

**(b.1)**

**(b.2)**
Ilustration of routing effects.

**Impact on streamflow metrics**

**(c.1)** Aggregated time step

**(c.2)**
Impact of routing configuration on performance metrics.

**Fluxes partitioning and parameters**

**(d.1)** Routing time step
Effects on flux partioning.

**(d.2)** Routing time step
Effects on the choice of parameter values.

**Flood frecuency and flow duration curves**

**(e.1)** Routing time step
Implications on frequency curves.

**(e.2)** Routing time step
Impacts on flow duration curves.

**Figure 2. Overview of the analysis framework used here. (a.1) VIC model simulations are conducted at hourly time steps for 3500 parameter sets, and (a.2) each runoff time series at each grid cell is aggregated to four additional time steps (2, 3, 4 and 6 h), and these new time series are routed with four schemes to produce 3500 (VIC parameters) x 5 (time steps) x 5 (Inst + four routing schemes) modeling configurations. (b) We illustrate routing effects on (b.1) simulated hydrographs during Fall 2008 and (b.2) simulated streamflow at various temporal resolutions. (c) For each configuration, we compute performance metrics and examine (c.1) the impacts of routing configuration on streamflow performance computed at various time steps, using the 1% best model runs, and (c.2) improvements in performance metrics across other gauge points in the Cautín River basin. (d.1) We analyze simulated mean annual water balance and baseflow contribution to total runoff, and (d.2) compare the best parameters sets for each configuration in terms of their normalized values. (e) Finally, we analyze the implications of routing configurations on (e.1) flood frequency and (e.2) flow durations curves.**

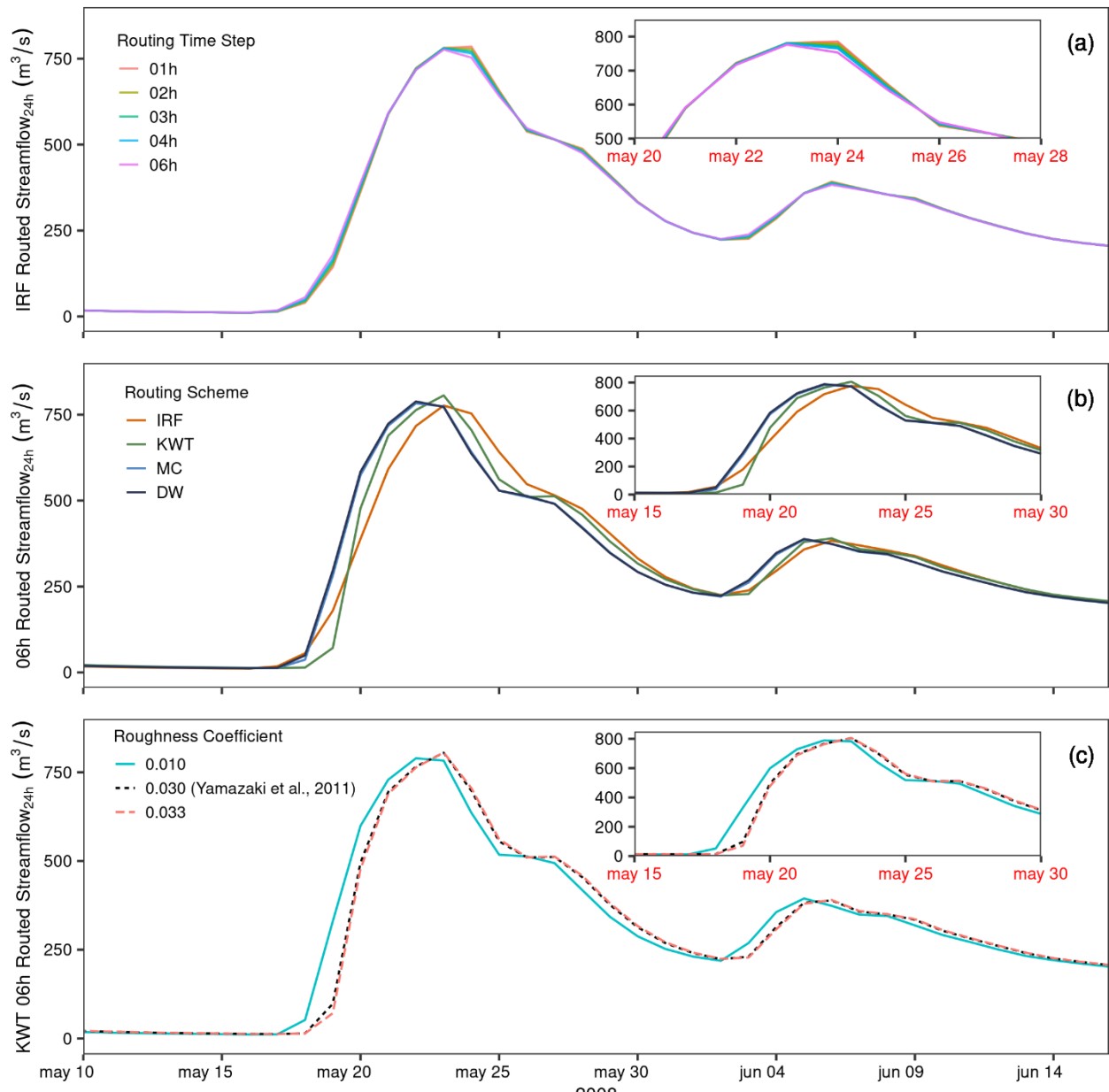

**Figure 3. Time series with daily streamflow at Cautín at Cajón, obtained from hourly VIC runoff outputs routed with different mizuRoute configurations. (a) Application of the Impulse Response Function (IRF) with five different routing time steps, (b) effects of different routing schemes using Δt = 6 h, and (c) effects of the Manning's roughness coefficient (*n*) when applying the kinematic wave routing scheme with Δt = 6 h (see text for details). In panel (c), the orange line (*n* = 0.033) is associated with the parameter set that maximizes the KGE computed with daily streamflow.**

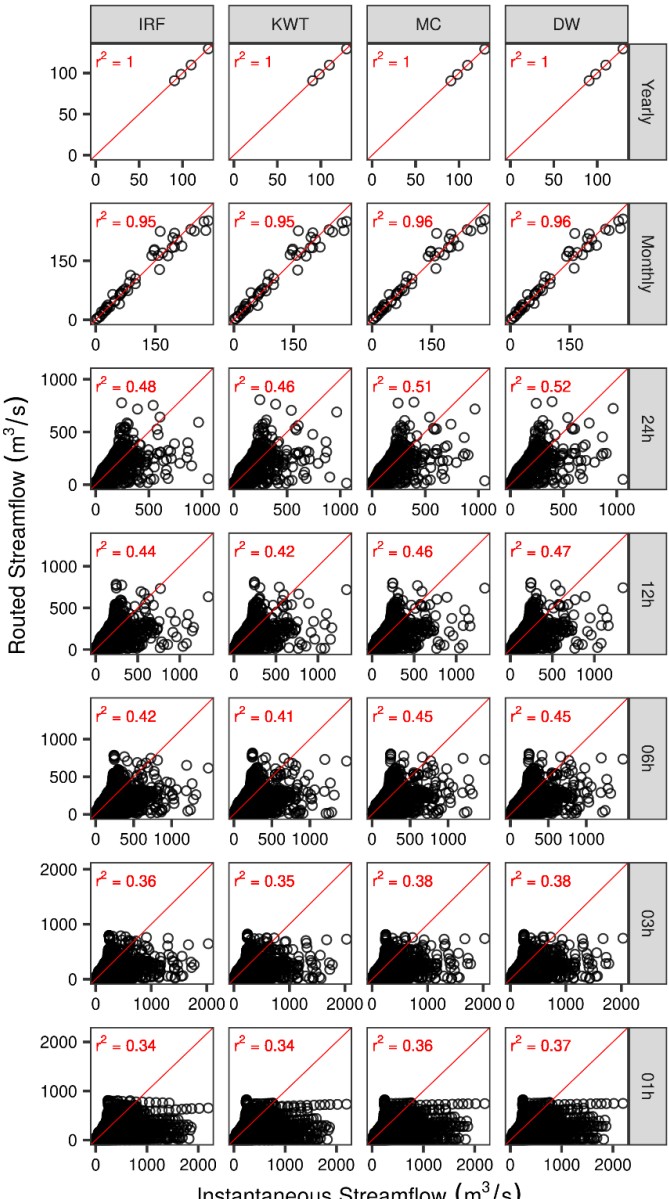

Figure 4. Simulated streamflow (VIC+mizuRoute) vs. instantaneous VIC runoff at Cautín at Cajón for the period April/2008-March/2012, using different time steps (rows) and routing schemes (columns): instantaneous runoff (Inst), Impulse Response Function (IRF), Kinematic Wave Tracking (KWT), Muskingum-Cunge (MC) and Diffusive Wave (DW). Mean yearly, monthly, daily and 12 h streamflows are computed from temporally averaged 6 h values, while 1 h, 3 h, and 6 h streamflows are obtained from mizuRoute simulations with 1 h, 3 h, and 6 h time steps, respectively. The 1:1 line is displayed in red with the coefficient of determination ($R^2$)

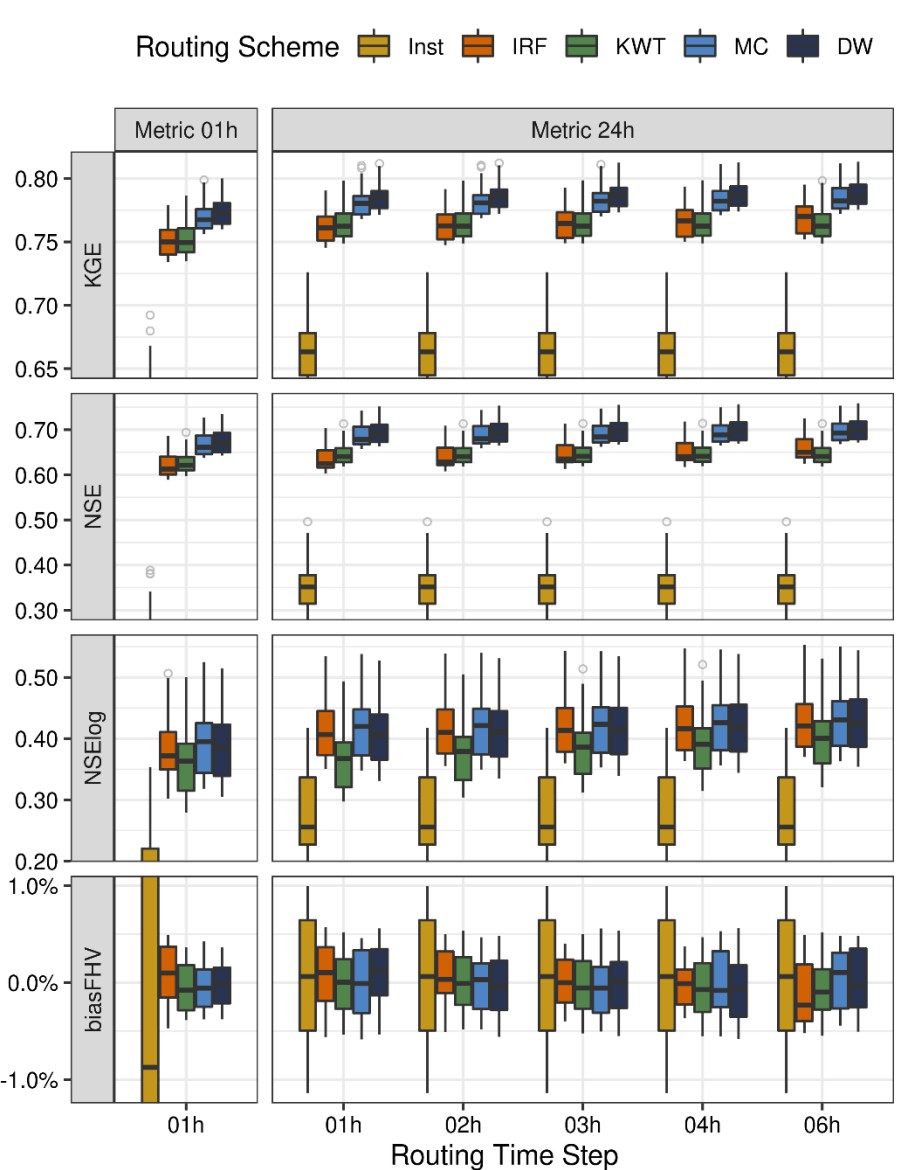

**Figure 5. Impact of routing scheme and routing time step on performance metrics (rows) for the period April/2008-March/2012 at Cautín at Cajón, computed with 1 h and 24 h discharge temporal resolutions (columns) and the 1% best parameter sets among those obtained through Latin Hypercube Sampling (see text for details). The results are presented for instantaneous runoff (Inst), Impulse Response Function (IRF), Kinematic Wave Tracking (KWT), Muskingum-Cunge (MC) and Diffusive Wave (DW).**

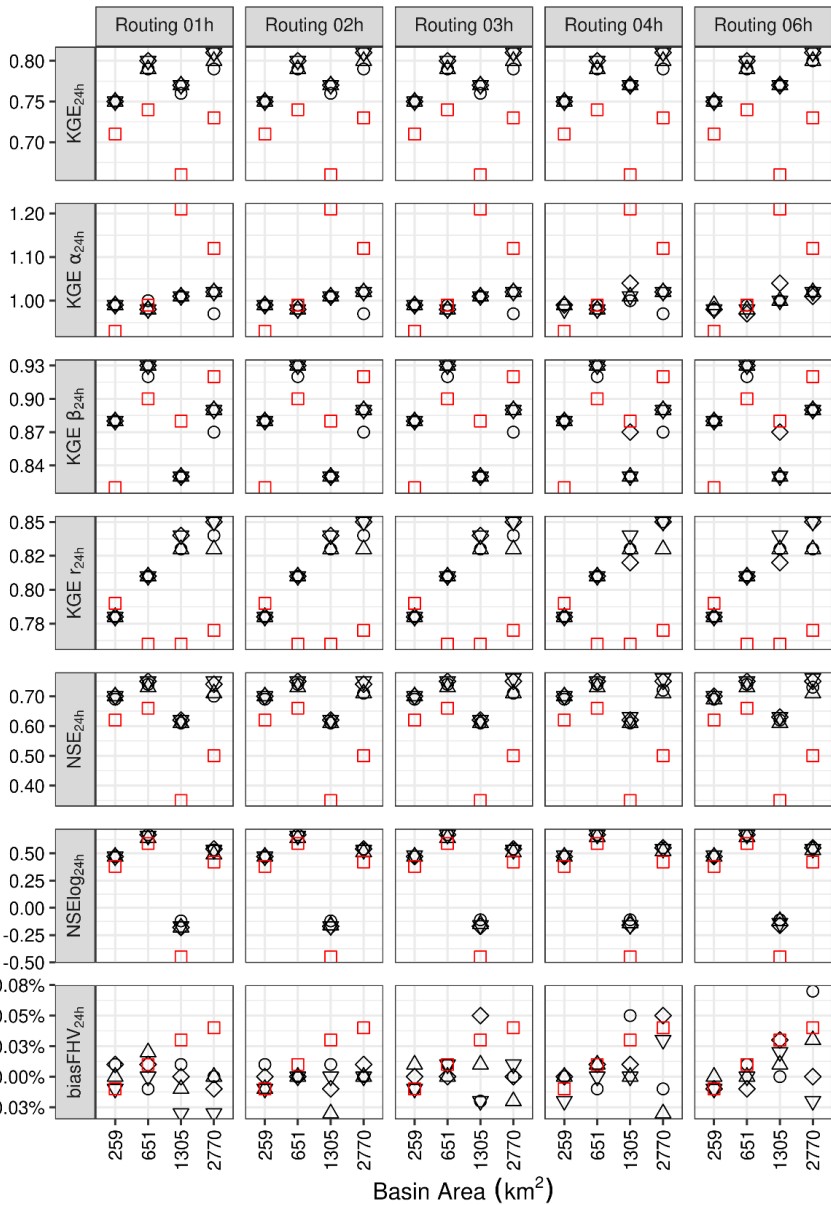

**Figure 6. Best metric value obtained with daily flows at each stream gage station for a given performance metric (rows), routing time step (columns), and routing scheme for the period April/2008-March/2012. For completeness, the Kling-Gupta Efficiency (KGE) components associated to the best KGE value are included. The results are presented for instantaneous runoff (Inst, red symbols), Impulse Response Function (IRF), Kinematic Wave Tracking (KWT), Muskingum-Cunge (MC) and Diffusive Wave (DW).**

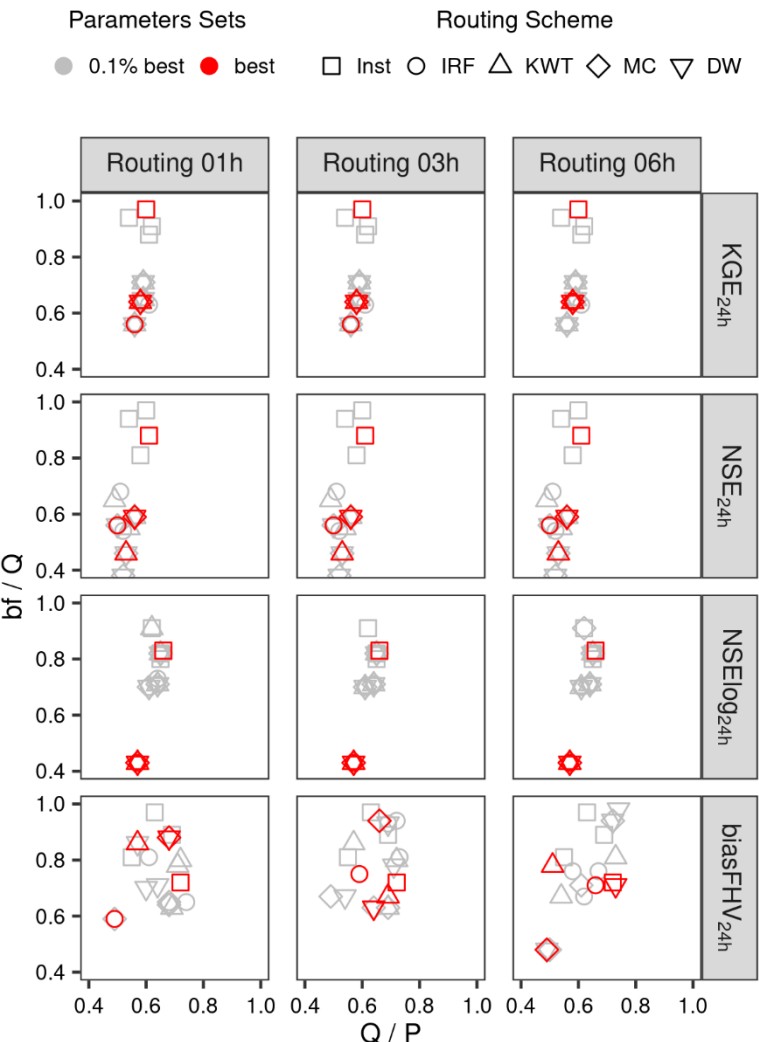

**Figure 7. Effects of performance metric (rows), routing time step (columns) and routing scheme on simulated mean annual water balance (characterized with the annual runoff ratio, x-axis) and the baseflow ratio (y-axis) obtained for the 0.1% best parameter sets at Cautín at Cajón (period April/2008-March/2012). The results are presented for instantaneous runoff (Inst), Impulse Response Function (IRF), Kinematic Wave Tracking (KWT), Muskingum-Cunge (MC) and Diffusive Wave (DW). In each panel, the results obtained with the parameter set (among the 3500 samples) that maximizes each metric are displayed in red, the results from a small ensemble (n = 4) with the best 0.1% VIC parameter sets are displayed in grey.**

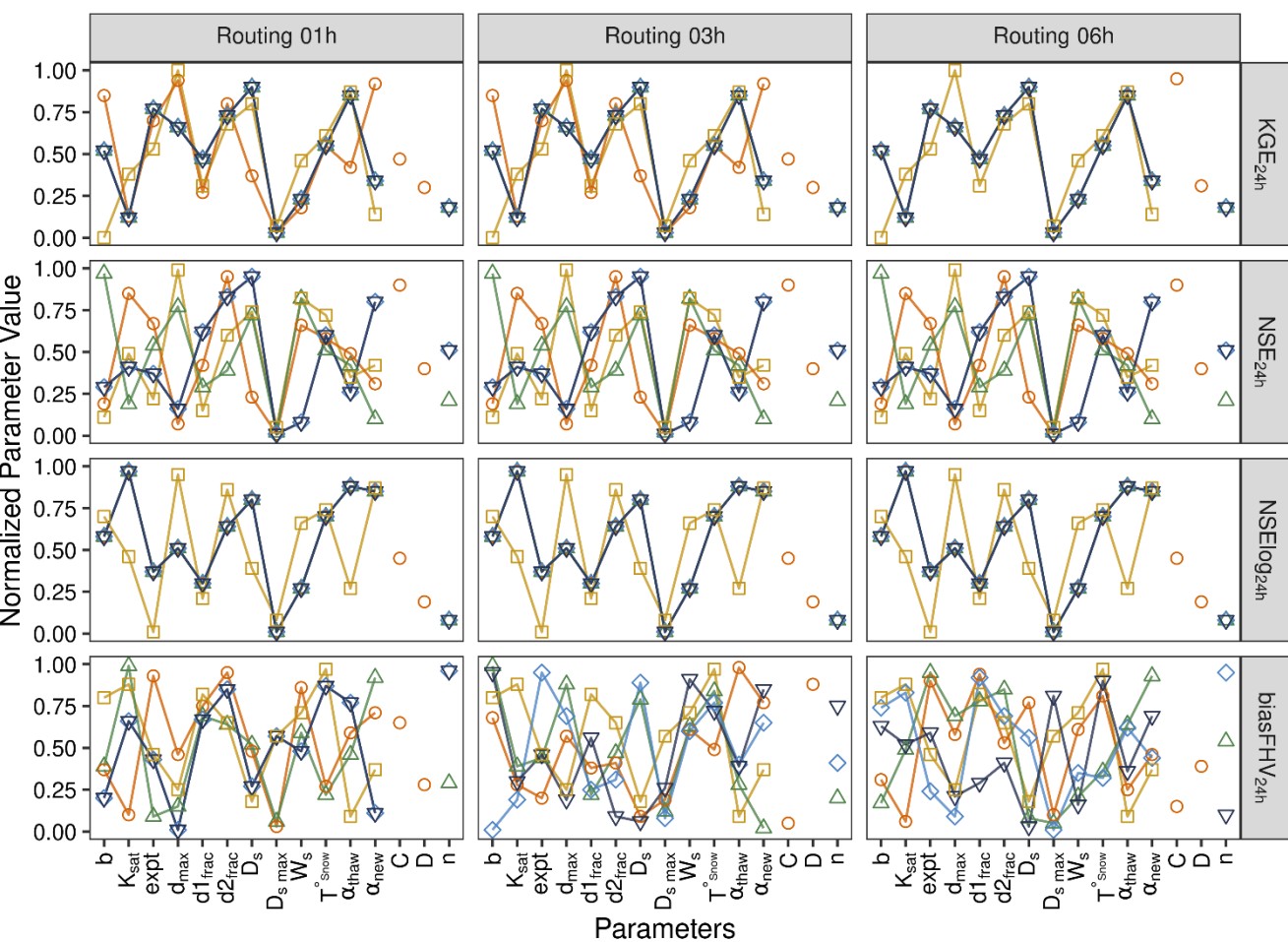

**Figure 8. Normalized parameter values for Cautín at Cajón associated to the best performance metric (period April/2008 – March/2012) obtained from the 3500 parameter sets produced with the Latin Hypercube Sampling, given a combination of routing scheme and routing time step. The symbols representing VIC parameters are linked with straight lines. The results are presented for daily instantaneous runoff (Inst), Impulse Response Function (IRF), Kinematic Wave Tracking (KWT), Muskingum-Cunge (MC) and Diffusive Wave (DW).**

835

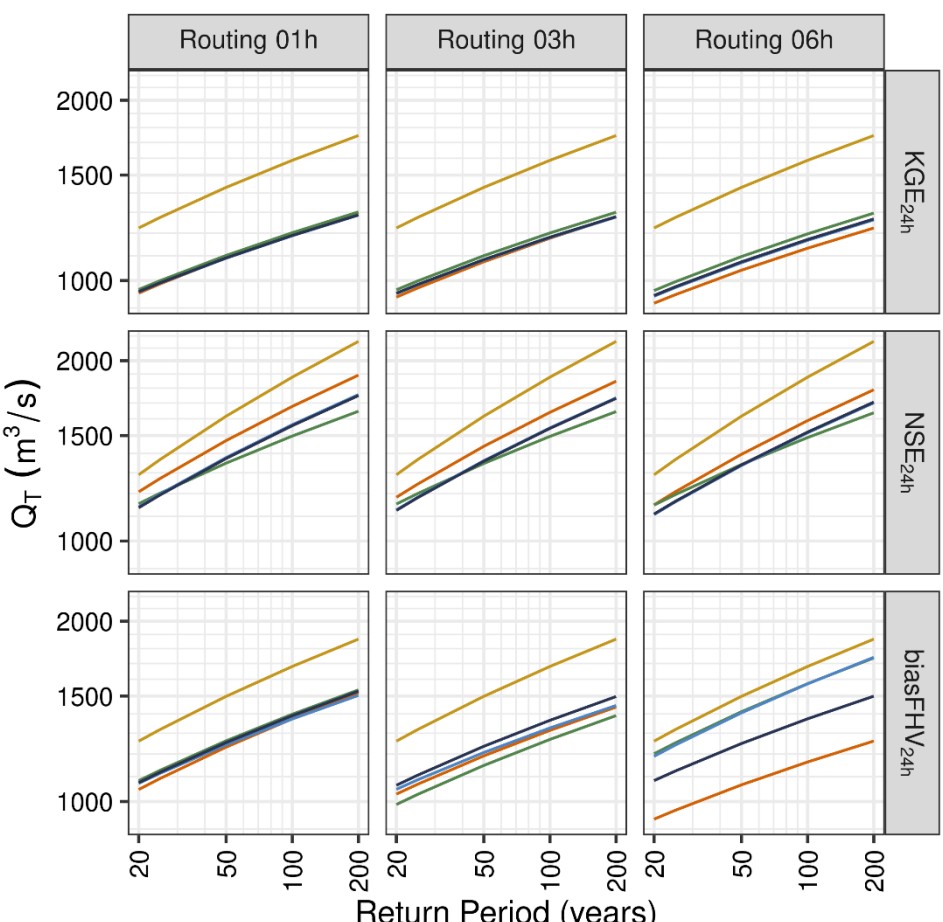

**Figure 9. Frequency curves for annual maximum daily flows (y-axis) in Cautín at Cajón, derived from numerical simulations conducted with different routing schemes, routing time steps (columns) and performance metrics (rows). All frequency curves are computed from annual time series of n = 35 annual maximum daily flows (April/1985 – March/2020) using a Log-Normal density function. The results are presented for instantaneous runoff (Inst), Impulse Response Function (IRF), Kinematic Wave Tracking (KWT), Muskingum-Cunge (MC) and Diffusive Wave (DW).**

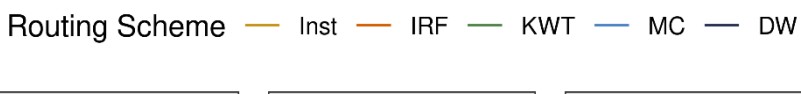

Figure 10. Mean daily flow duration curves for the period April/1985 – March/2020 in Cautín at Cajón derived from different routing schemes, routing time steps (columns) and performance metrics (rows). The results are presented for daily instantaneous runoff (Inst), Impulse Response Function (IRF), Kinematic Wave Tracking (KWT), Muskingum-Cunge (MC) and Diffusive Wave (DW) routing schemes.