# Peer review of "To what extent does river routing matter in hydrological modeling?"

_Hydrology and Earth System Sciences, 2022_

## Author Comment (AC1)

Replies to reviews

**"To what extent does river routing matter in hydrological modelling?"**

Nicolás Cortés-Salazar, Nicolás Vásquez, Naoki Mizukami, Pablo A. Mendoza, and Ximena Vargas

We provide responses to each individual point below. For clarity, comments are given in italics, and our responses are given in plain blue text.

**Anonymous Referee #1**

*The authors conducted a very comprehensive sensitivity analyses of the effects of adding an additional river routing model with various schemes to a hydrological model. The publication shows promise as a great reference work for model experiment setup. In general, the publication is well written and the arguments for conducting the study are clear. The decisions made regarding the methods are well-argued (with the exception of 1) and the results are valuable for the hydrologic community. The limitation of this study are well described. It is understandable given the scope of the study and the data requirements that the authors evaluated a single catchment. For future research, I am eager to discover how the results of this study would be different in a more gentle sloping catchment or for various catchment sizes (e.g using CAMELS-CH Alvarez-Garreton et al., 2018).*

We thank the referee for meticulously reviewing our manuscript and providing several constructive suggestions. We are especially grateful for the referee's positive feedback.

*That being said, the publications needs some extra work. The main points that need attention are argumentation for hydrological model aggregation, the structure of text and figures, additional reflection on the meaning of study results, and the archiving of code and data.*

In this document, we provide our detailed responses and also mention how we plan to address the reviewer's comments in a future version of this manuscript.

*Major comments:*

*Temporal aggregation of hydrological model results*

*In section 3.3 the authors state that for each parameter set the VIC model is run at hourly time-steps and the results are temporally aggregated to various coarser time-steps. In my opinion this is an assumption that there are no non-linear processes in time within the hydrological models. The necessity for this assumption is clear as it results in a clean model experiment. However, the authors should more clearly state this assumption and reflect on this in section 5.1 (last paragraph) and 5.2. I'm curious to read the authors response.*

This is a good point, and we fully agree with this reviewer that this limitation should be made clear. We have added the following text to make explicit the assumption that the reviewer refers to in section 4.1:

"It should be noted that, in this step, we assume the absence of non-linear processes in time within the hydrological model, so hourly VIC runoff can be temporally aggregated to coarser time steps."

We have added the following sentence in Section 6.1 (discussion section):
"Accordingly, variations in the VIC model time step – which is fixed to $\Delta t = 1$ h here – may also alter the selection of parameters and performance measures (see section 5.2)."

We will also add the following text to section 6.2 (discussions section):

"Another important assumption is the lack of non-linear processes in time within the hydrological model, in order to aggregate hourly runoff to coarser time steps. Such decision was required to isolate the impact of hydrologic model configuration from river routing decisions, and achieve a clean experimental design, though we recognize that the choice of hydrological model time step may also alter performance metrics (e.g., Bruneau et al., 1995; Wang et al., 2009)."

*Structure of text*

*The authors conducted a lot of analyses which to their credit lead to an abundance of methodology steps and results. This makes section 3.5 difficult to read and therefore it needs restructuring. I suggest to use numbering to make the steps more clear even if this disrupts the flow of the text.*
*What might also help the reader is a model run results matrix in the form of a Table that uses the same numbering. This makes it clearer for the reader what results can be expected for each type of model run configuration.*

We plan to divide the original section 3 into two sections: (3) Models (which describes VIC and mizuRoute), and (4) Experimental setup. The latter section starts by numbering the main steps and analyses conducted, followed by details descriptions for (4.1) parameter sampling and streamflow simulations, (4.2) objective functions, and (4.3) flood frequency analyses. We hope that the proposed restructuring of section 3 will clarify the approach used here.

*Structure of figures*

*There are issues with the presentation of the results in the figures. Overall the image quality (dpi) per figure needs to be higher. The colours used to represent the individual routing schemes are inconsistent, please check all figures.*

We will improve the image resolution and revised the colours used for the individual routing schemes.

*Figure 1: It is difficult to find the catchment on the left panel (1a). Outlining the catchment in red and using a softer tone for the country would help. The colours for elevation bands in 1b are difficult to distinguish, similar issue with the sub-basins in 1c.*

We will modify Figure 1 to include the reviewer's recommendation. We will change the colour for Continental Chile, and will highlight the basin boundaries. Regarding panel b), the colour scale will change, and, in panel c), we will improve the contrast between streams and sub-basin delineation.

*Figure 2: Increase image quality.*

We will improve the image resolution, following the reviewer's recommendation.

*Figure 3: Highlighting the horizontal axes in red would help find the period of the zoom boxes.*

We will highlight the horizontal axis in red for the period displayed in the zoom boxes.

*Figure 5: Colours are difficult to distinguish, suggest using the same colors for each scheme as in Figure 3. The vertical axes of each column varies, ticks for KGE are in steps of 0.2 while those of NSE are 0.4. This makes it nearly impossible to assess the relative differences in objective functions. I suggest using the same tick sizes with the exception of NSElog.*

We will modify the colours and try different tick sizes, following the reviewer's suggestion.

*Figure 6: There is almost no reference to the different basin areas that are shown using the horizontal axis. It would make the figure a lot clearer if only the 2770 basin area was shown and the individual schemes were plotted next to each other. I suggest placing the results for the other basin areas in the appendix.*

We will re-structure Figure 6 to keep only the basin outlet, and show the rest of the basins in Supplements.

*Figure 7: Similar to Figure 6 this figure is difficult to read. The total width of the horizontal axis does not add information, therefore I suggest to make the ticks smaller.*

Our original aim was to use the same limits for x and y-axes to make clear that routing has a larger impact on the baseflow fraction, compared to the mean annual runoff ratio. In any case, we will try alternative resolutions for the x-axis, as the reviewer suggests.

*Figure 8: Increase the image quality. I suggest to make a separate table for the objective function results.*

We will increase the image quality and will include the results referred to in a separate table.

*Reflection on the meaning of study results*

*The discussion section 5.1 can be extended by reflecting more on the implications of results. For example, we understand what is happening to the hydrological model in the absence of river routing. Compensation through baseflow and no considerable change in precipitation, evapotranspiration and runoff partitioning. What is missing is, what the implication are for users and why it is important to get these parts right in hydrological model setups. This is also the case for the results in 4.4.*

To incorporate the reviewer's suggestion, we plan to expand the second paragraph in section 6.1 (discussion section):

"The results presented here show that the implementation and configuration of river routing schemes are also relevant for medium and low flows. For example, including river routing provided higher values for NSE-log (Figures 5 and 6) – improving the simulation of low discharges – and modified the shape of the mid and low flow segments in the FDC (Figure 10), which are characteristic signatures of 'flashiness' in runoff response and long term baseflow, respectively (Yilmaz et al., 2008). The effects of river routing are also reflected in the partitioning of total runoff between baseflow and surface runoff. In fact, the results presented here show that the parameter search process compensates for the lack of routing by modifying other fluxes and state variables (Khatami et al., 2019) to increase streamflow-oriented performance metrics. In our case, the contribution of baseflow to total runoff increases by >20% when river routing is excluded, which is achieved by modifying soil parameters –especially $W_s$, one of the most sensitive for baseflow processes (Sepúlveda et al., 2022) – to delay the streamflow response. This result suggests that including routing processes may impact the outcomes from drought-oriented studies, since baseflow is the primary flux sustaining streamflow during water scarcity periods (Karki et al., 2021).

Conversely, we did not find considerable variations in the partitioning of precipitation between evapotranspiration and runoff in the absence of river routing (Figure 10), which means that this process is relatively less important for hydroclimatic analysis at the annual time scale."

*In addition, the selection of objective-function is discussed but there is no discussion on multi-objective calibration and how these might affect the results. There is reflection needed on the relevance of the differences in objective-function values. What does a difference of xx KGE mean?*

In response to the reviewer's observation, we plan to add the following paragraph in section 6.2 (discussion section):

"Finally, we only considered a single-objective (e.g., NSE, KGE) parameter search based on a Monte Carlo sampling scheme. Future studies could characterize the impacts of river routing schemes exploiting single-objective optimization algorithms (e.g., Duan et al., 1992; Tolson & Shoemaker, 2007), or address multi-objective problems using Pareto principles (e.g., Yapo et al., 1998; Pokhrel et al., 2012; Shafii & Tolson, 2015). Although different behavioral parameter sets and, therefore, different internal fluxes could be obtained, we hypothesize that similar conclusions could be drawn regarding the benefits of river routing representation to achieve realistic streamflow simulations. Nevertheless, further research is needed to understand implications for catchments with different hydroclimatic regimes and physiographic characteristics."

*Data*

*The authors state "The codes used in this study are available from the corresponding authors upon reasonable request". What does reasonable mean?*

*The Copernicus data policy (https://publications.copernicus.org/services/data_policy.html) states "In addition, data sets, model code, video supplements, video abstracts, International Geo Sample Numbers, and other digital assets should be linked to the article through DOIs in the assets tab."*

*In the spirit of open-science I strongly encourage the authors to do so. I leave it up to the editor to determine whether this is a requirement for publication.*

We will create a repository on Zenodo to make our code, data and results publicly available:

Cortés-Salazar, Nicolás; Vásquez, Nicolás; Mizukami, Naoki; Mendoza, Pablo; Vargas, Ximena. (2023). Hydrology and river routing models for the Cautin River basin, Araucania Region, Chile [Data set]. Zenodo. https://doi.org/ 10.5281/zenodo.7582302

*Minor comments:*

*Refrain from using acronyms in figure captions. The style of figure captions is inconsistent, e.g. use of ":", or ";", or ","*

We revise all figure captions in the manuscript, and have harmonized the writing style. We have also decided to keep the acronyms in the figure captions for consistency with the figures and the text; however, we have spelled them out to facilitate the reading.

*Lines 71-72: SWAT model is missing a reference.*

We specify the acronym for SWAT and added a reference (Arnold et al., 1998), following the reviewer's recommendation.

*Lines 76 -80: Very long sentence, needs restructuring.*

We will re-word this sentence as follows:

"Although many past studies have shown that the choice of routing scheme affects streamflow simulations, efforts for improving their accuracy have been made by configuring hydrologic model and routing model independently. Hydrologists still focus on parameter calibration to improve discharge simulations, neglecting the potential impacts of river routing configuration, especially routing scheme and time step (Beck et al., 2020; Newman et al., 2021). On the other hand, routing

model evaluation uses hydrologic model output, which contains varying degree of errors, making it difficult to evaluate routing models especially for basin or greater spatial domain (e.g., Mizukami et al., 2016; F. Zhao et al., 2017), and often use synthetic river discharge (Price, 2009; David et al., 2011)."

*Line 93: remove "apparently"*

We  will remove this word, following the reviewer's suggestion.

*Lines 251 – 254: This is a bold claim that I would remove as it does not add value to speculate.*

We will remove this sentence, following the reviewer's recommendation.

*Line 349: "MC approach", change to machine learning approach.*

MC stands for Muskingum-Cunge. The acronym is defined in section 3.2, but here we decide to spell out to avoid confusion among readers. Thanks for this observation!

*Personal dislike of the use of the word "indeed" throughout the publication.*

We will remove the word 'indeed' from the manuscript, following the reviewer's recommendation.

**References**

Arnold, J. G., Srinivasan, R., Muttiah, R. S., & Williams, J. R. (1998). Large area hydrologic modeling and assessment part I: model development. *Journal of the American Water Resources Association*, *34*(1), 73–89. https://doi.org/10.1111/j.1752-1688.1998.tb05961.x

Beck, H. E., Pan, M., Lin, P., Seibert, J., van Dijk, A. I. J. M., & Wood, E. F. (2020). Global Fully Distributed Parameter Regionalization Based on Observed Streamflow From 4,229 Headwater Catchments. *Journal of Geophysical Research: Atmospheres*, *125*(17). https://doi.org/10.1029/2019JD031485

Bruneau, P., Gascuel-Odoux, C., Robin, P., Merot, P., & Beven, K. (1995). Sensitivity to space and time resolution of a hydrological model using digital elevation data. *Hydrological Processes*, *9*(1), 69–81. https://doi.org/10.1002/hyp.3360090107

David, C. H., Maidment, D. R., Niu, G. Y., Yang, Z. L., Habets, F., & Eijkhout, V. (2011). River network routing on the NHDPlus dataset. *Journal of Hydrometeorology*, *12*(5), 913–934. https://doi.org/10.1175/2011JHM1345.1

Duan, Q., Sorooshian, S., & Gupta, V. (1992). Effective and Efficient Global Optimization for Conceptual Rainfal-Runoff Models. *Water Resources Research*, *28*(4), 1015–1031.

Karki, R., Krienert, J. M., Hong, M., & Steward, D. R. (2021). Evaluating Baseflow Simulation in the National Water Model: A Case Study in the Northern High Plains Region, USA. *Journal of the American Water Resources Association*, *57*(2), 267–280. https://doi.org/10.1111/1752-1688.12911

Khatami, S., Peel, M. C., Peterson, T. J., & Western, A. W. (2019). Equifinality and Flux Mapping: A New Approach to Model Evaluation and Process Representation Under Uncertainty. *Water Resources Research*, *55*(11), 8922–8941. https://doi.org/10.1029/2018WR023750

Mizukami, N., Clark, M. P., Sampson, K., Nijssen, B., Mao, Y., McMillan, H., et al. (2016). mizuRoute version 1: a river network routing tool for a continental domain water resources applications. *Geoscientific Model Development*, *9*(6), 2223–2238. https://doi.org/10.5194/gmd-9-2223-2016

Newman, A. J., Stone, A. G., Saharia, M., Holman, K. D., Addor, N., & Clark, M. P. (2021). Identifying sensitivities in flood frequency analyses using a stochastic hydrologic modeling system. *Hydrology and Earth System Sciences*, *25*(10), 5603–5621.

https://doi.org/10.5194/hess-25-5603-2021

Pokhrel, P., Yilmaz, K. K., & Gupta, H. V. (2012). Multiple-criteria calibration of a distributed watershed model using spatial regularization and response signatures. *Journal of Hydrology*, *418–419*, 49–60. https://doi.org/10.1016/j.jhydrol.2008.12.004

Price, R. K. (2009). An optimized routing model for flood forecasting. *Water Resources Research*, *45*(2), 1–15. https://doi.org/10.1029/2008WR007103

Sepúlveda, U. M., Mendoza, P. A., Mizukami, N., & Newman, A. J. (2022). Revisiting parameter sensitivities in the variable infiltration capacity model across a hydroclimatic gradient. *Hydrology and Earth System Sciences*, *26*(13), 3419–3445. https://doi.org/10.5194/hess-26-3419-2022

Shafii, M., & Tolson, B. A. (2015). Optimizing hydrological consistency by incorporating hydrological signatures into model calibration objectives. *Water Resources Research*, *51*(5), 3796–3814. https://doi.org/10.1002/2014WR016520

Tolson, B. A., & Shoemaker, C. A. (2007). Dynamically dimensioned search algorithm for computationally efficient watershed model calibration. *Water Resources Research*, *43*(1), 1–16. https://doi.org/10.1029/2005WR004723

Wang, Y., He, B., & Takase, K. (2009). Effects of temporal resolution on hydrological model parameters and its impact on prediction of river discharge. *Hydrological Sciences Journal*, *54*(5), 886–898. https://doi.org/10.1623/hysj.54.5.886

Yapo, P. O., Gupta, H. V., & Sorooshian, S. (1998). Multi-objective global optimization for hydrologic models. *Journal of Hydrology*, *204*(1–4), 83–97. https://doi.org/10.1016/S0022-1694(97)00107-8

Yilmaz, K. K., Gupta, H. V., & Wagener, T. (2008). A process-based diagnostic approach to model evaluation: Application to the NWS distributed hydrologic model. *Water Resources Research*, *44*(9), W09417. https://doi.org/10.1029/2007WR006716

Zhao, F., Veldkamp, T. I. E., Frieler, K., Schewe, J., Ostberg, S., Willner, S., et al. (2017). The critical role of the routing scheme in simulating peak river discharge in global hydrological models. *Environmental Research Letters*, *12*(7). https://doi.org/10.1088/1748-9326/aa7250

---

## Author Comment (AC2)

Replies to reviews

**"To what extent does river routing matter in hydrological modelling?"**

Nicolás Cortés-Salazar, Nicolás Vásquez, Naoki Mizukami, Pablo A. Mendoza, and Ximena Vargas

We provide responses to each individual point below. For clarity, comments are given in italics, and our responses are given in plain blue text.

**Anonymous Referee #2**

*I have finished my review of the paper "To what extent does river routing matter in hydrological modeling", by Cortés-Salazar et al., submitted to HESS. This generally well-written paper attempts to examine the influence of routing algorithm and time step on model performance using subsets of 3500 different runoff regimes generated using the VIC hydrological model. Very mild differences were found between algorithm and time step choices, with one exception: where routing was not simulated, the performance was consistently poor relative to models which simulate routing. Unfortunately, this is not a very compelling result.*

*While the approach was rigorous in the sense that it compiled data from thousands of model simulations, it suffers from a number of critical methodological issues.*

We thank the referee for his/her time on reviewing our manuscript and providing several constructive suggestions. In this document, we provide our detailed responses and also mention how we plan to address the reviewer's comments in a future version of this manuscript.

*I discuss a few of these major issues below:*

*Routing is most influential on peak magnitude and timing of large events; both of these are poorly captured by integrated hydrograph metrics such as KGE and NSE. Peak flow differences after calibrating the routing models would be a much more useful metric for evaluating routing model performance. By using integrated measures such as KGE, the critical differences between routing algorithms are not discernable (as seen in nearly all of the reported results).*

We agree with this reviewer that including additional high-flow related metrics would increase the impact of this work. Therefore, we will add the annual peak flow bias (Mizukami et al., 2019) and the percent bias in the high flow segment of the flow duration curve (Yilmaz et al., 2008) as calibration metrics. The justification for including these metrics and related equations will be included in section 4.2 ("Objective functions") of the revised manuscript.

In this work, we decide to include KGE, NSE and log-NSE in the analysis, since these integrated metrics are not only of interest for the hydrologic modeling community – especially for parameter calibration and evaluation (e.g., Fowler et al., 2018; Knoben et al., 2019; Clark et al., 2021) –, but also for the river routing community. Indeed, several examples of river routing scheme assessments can be found using the KGE (e.g., Pereira et al., 2017; Hoch et al., 2019; Qiao et al., 2019; Thober et al., 2019; Munier & Decharme, 2022), NSE (e.g., Yamazaki et al., 2011; Ye et al., 2013; Zhao et al., 2017; ElSaadani et al., 2018; Nguyen-Quang et al., 2018; Fleischmann et al., 2019, 2020) and even the NSE using flows in logarithmic space (Paiva et al., 2013). We believe that the joint analysis of traditional streamflow performance metrics and high flow metrics will expand the target audience of this work. We will provide a proper justification for selecting these metrics in the new section 4 ("Experimental setup") of the revised manuscript.

*Each of the figures in the report are reporting ALL of the output from the simulations, regardless of whether it is important or interpretable or worthy of interpretation.*

Figure 5 is the only one that contains results from all the parameter samples, though we will modify this to address the following reviewer's comment (see next response).

*For instance, figure 5 reports KGE, NSE, and NSE of log transformed flows for all 3500 simulations with multiple timesteps, multiple routing schemes. In addition to the only interpretable result from this figure is that no routing is outperformed by routing, there is little utility in comparing NSE values of 0.2-0.3 (the approximate median of these simulations) -differences in NSE below about 0.5 are nearly arbitrary in that a hydrograph with an NSE of 0.2 may not be visibly preferable to an NSE of 0.05. The only feature of this plot referred to in the text was the maximum metric value. Why not simply report that?*

The original motivation of Figure 5 was to illustrate the impact that incorporating river routing modelling may have on streamflow performance metrics across the VIC parameter space. Nevertheless, we fully agree that it would make more sense to simply illustrate the results for a subset of behavioural parameter sets (as we did in Figure 7). Hence, in the revised manuscript we select and report only the best 1% of runs, following the approach of Melsen et al. (2016).

*Critically, because the parameters of the VIC model are arbitrary, the comparisons of even the best models are in effect the results of Monte Carlo calibration, the least efficient optimization approach.*

We decide to use a Monte Carlo parameter sampling approach because, rather than seeking for an optimal parameter set, our primary goal is to assess the impacts of different river routing configurations on streamflow metrics across the model parameter space. In particular, Latin Hypercube Sampling is a common strategy to sample the parameter space and identify behavioural parameter sets for a specific target metric (e.g., Andréassian et al., 2014; Broderick et al., 2016; Melsen et al., 2016, 2019; Guse et al., 2017; Khatami et al., 2019). We will clarify this in section 4.1 ("Parameter sampling and streamflow simulations") of the revised manuscript, adding the following text:
"Since we aim to examine the impacts of different routing schemes on streamflow performance metrics across the parameter space, rather than seeking for an optimal parameter set, we use the Latin Hypercube Sampling (LHS) method, which is a common strategy to sample the parameter space and identify behavioral sets for specific target metrics (e.g., Andréassian et al., 2014; Broderick et al., 2016; Melsen et al., 2016, 2019; Guse et al., 2017; Khatami et al., 2019)."

*Comparing the 'best' models when these are not rigorously determined to be the actual best for each algorithm (rather than a sampling error) is problematic. For this comparison to be rigorous, I don't see how to do this without simultaneous calibration of routing and land surface parameters, an issue the authors acknowledge in section 5.2.*

River routing parameters were excluded from the original setup in order to make a clean numerical experiment (recall that the baseline model has no routing module). However, we appreciate the reviewer's concern, and in response to this comment we are conducting new LHS experiments that include river routing parameters (see response to the following comment).

*In practice, the routing parameters (such as Manning's n) would be calibrated in conjunction with VIC model parameters, likely further diminishing any incremental performance differences between the routing models.*

In order to address this point, we have repeated the parameter sampling experiment, including 13 VIC parameters and the routing parameters: one (the Manning roughness coefficient) for the Kinematic Wave, Diffusive Wave and Muskingum-Cunge algorithms, and two for the Impulse Response Function method (see Table S1). The new results are displayed in Figure S1.

Table S1. Model parameters sampled in this study.

| Parameter | Units | Lower value | Upper value | Description |
|---|---|---|---|---|
| Infilt | - | 0.01 | 0.99 | Variable infiltration curve parameter |
| $D_s$ | - | 0.1 | 0.9 | Fraction of $D_{s\,max}$ where non-linear baseflow occurs |
| $D_{s\,max}$ | mm/d | 0.1 | 300 | Maximum velocity of baseflow |
| $W_s$ | - | 0.1 | 0.9 | Fraction of maximum soil moisture where non-linear baseflow occurs |
| expt | - | 3.1 | 10 | Exponent of Campbell's equation for hydraulic conductivity |
| $d_{max}$ $d_1$ $d_2$ $d_3$ | m | 0.5 $0.05\,d_{max}$ $0.21\,d_{max}$ $0.74\,d_{max}$ | 5 $0.2\,d_{max}$ $0.7\,d_{max}$ $0.1\,d_{max}$ | Depth of soil layers 1, 2 and 3 |
| $K_{sat}$ | mm/d | 1 | 1000 | Saturated hydraulic conductivity |
| $T_{max,snow}$ | (°C) | -10 | 10 | Maximum temperature for snowfall |
| $\alpha_{thaw}$ | - | 0.75 | 0.90 | Decay of albedo |
| $\alpha_{new}$ | - | 0.85 | 0.95 | Maximum albedo for fresh snow |
| $n$ | $s/m^{1/3}$ | 0.024 | 0.075 | Roughness coefficient of Manning (Barnes, 1967) |
| $C$ | $m/s$ | 0.25 | 10 | Advection coefficient (Allen et al., 2018) |
| $D$ | $m^2/s$ | 200 | 4000 | Diffusion coefficient (Melsen et al., 2016) |

[Figure]

Figure S1. New parameter sampling results for the Cautín at Cajón River basin. Impact of routing scheme and routing time step on performance metrics (rows) computed for the period April/2008-March/2012, using different discharge temporal resolutions for metric calculation (columns) across a large sample of VIC and routing parameter sets obtained through Latin Hypercube Sampling. The results are presented for instantaneous runoff (Inst), Impulse Response Function (IRF), Kinematic Wave Tracking (KWT), Muskingum-Cunge (MC) and Diffusive Wave (DW), and the metrics are the Kling-Gupta Efficiency (KGE), the Nash-Sutcliffe efficiency in both raw (NSE) and logarithmic (NSElog) space, the annual peak flow bias (APFB) and the percent bias in the high flow segment of the flow duration curve (bias FHV). Each boxplot shows, for each configuration, the best 1% from 3500 parameter sets.

As seen in Figure S1, the benefits of representing river routing processes are not only limited to popular performance measures like KGE or NSE, but also to metrics that emphasize low flow simulations (NSE-log), and metrics for high flow applications. We plan to update all the figures of the revised manuscript based on the new experimental setup.

*The fundamental results discussed here are obvious without the testing herein.*

We provide individual responses to all the reviewer's comments on this matter below:

*Routing is better than no routing.*

Of course, the incorporation of as many hydrological processes as observed for the natural system of interest – including river routing – is desirable to achieve "fidelius" (Gharari et al., 2021) simulations in hydrology and land surface models. Nevertheless, the degree of improvement for application-specific metric is not obvious, considering that the interplay between model structural uncertainty (here, provided by different routing schemes) and parametric uncertainty is not fully understood and, therefore, it is a topic of active research (e.g., Günther et al., 2020; e.g., Pilz et al., 2020; Spieler et al., 2020; Chlumsky et al., 2021; Zhou et al., 2023). In particular, it is not clear to which degree the perturbation of hydrological model parameters can compensate for the lack of river routing representation – a gap that this work intends to fill and is now highlighted in the introduction.

*Low flows are not as impacted by routing differences.*

In principle, the effects of a specific model configuration (i.e., routing scheme or routing time step) on certain processes (e.g., high or low flows) are not comparable, since these are assessed with different performance metrics (e.g., Mizukami et al., 2019; Zhou et al., 2023). Additionally, the results presented here show that the implementation and configuration of river routing schemes are also relevant for medium and low flows. For example, including river routing provided higher NSE-log values (Figures 5 and 6 in the original submission) – improving the simulation of low flows – and modified the shape of the mid and low flow segments in the FDC (Figure 10 in the original submission), which are characteristic signatures of 'flashiness' in runoff response and long term baseflow, respectively (Yilmaz et al., 2008). The effects of river routing are also reflected in the partitioning of total runoff between baseflow and surface runoff. In fact, the results presented here show that the parameter search process compensates for the lack of routing by modifying other fluxes and state variables (Khatami et al., 2019) to increase streamflow-oriented performance metrics. In our case, the contribution of baseflow to total runoff increases by >20% when river routing is excluded, which is achieved by modifying soil parameters –especially $W_s$, one of the most sensitive for baseflow processes (Sepúlveda et al., 2022) – to delay the streamflow response. This result suggests that including routing processes may impact the outcomes from drought-oriented studies, since baseflow is the primary flux sustaining streamflow during water scarcity periods (Karki et al., 2021). All these points are now highlighted in the discussion (Section 6.1).

*Parameter compensation occurs in hydrological models.*

We agree that this is a well-known issue in the hydrologic modelling community, including compensation of hydrologic model parameters on forcing errors (e.g., Baez-Villanueva et al., 2021; Wang et al., 2023) and hydrologic model structure deficiencies (e.g., Saavedra et al., 2022). To the best of our knowledge, however, this is the first study that characterizes how hydrologic model parameter estimation can compensate for the absence of river routing representation, with focus on performance metrics, parameter values, flux partitioning and signatures used for water resources applications (in our case, flood frequency and flow duration curves). Our results show that, in this case, parameter compensation is not trivial, and depends on the calibration metric and fluxes analysed (Figure 7).

*Routing impacts are not visible when averaged over a monthly time step.*

The results presented here demonstrate exactly the opposite. Indeed, the second row in Figure 4 (original submission) shows that the difference in monthly flows between routed vs. non-routed flows can be as large as 63.2 m$^3$/s. We thank the reviewer for this comment, and we will modify the text in order to clarify this point:

"At monthly time steps, the differences between routed and instantaneous runoff reduce considerably, although these still can be as large as 63.2 m$^3$/s (i.e., a 29.3% difference using routed runoff as the reference)."

*Models with instantaneous routing (i.e., no routing) have higher flows.*

We agree (and also shown in Figures 9 and 10) that excluding routing yields higher flows. Obvious errors observed in the simulations from a model without routing are actually timing errors (due to the absence of travel time in the channel) and variability error (due to no attenuation accounted for). These types of errors are often assumed to become negligible when temporally aggregating non-routed flows. Nevertheless, this study unveils that the impacts of river routing go beyond the simulation of high flows at sub-daily or daily time steps, affecting streamflow simulations even at the monthly time scale (see previous response). The extent to which these effects propagate across modelling decisions is not trivial and has not been previously documented.

*As is, I do not see an additional contribution from this work beyond that which exists in the literature. It is for the above reasons that I must recommend rejection.*

We regret that the reviewer failed to agree with us regarding the contribution of the manuscript. We will address all the critiques raised in this review, and we will convey more clearly the relevance of our work in the revised version.

**References**

Andréassian, V., Bourgin, F., Oudin, L., Mathevet, T., Perrin, C., Lerat, J., et al. (2014). Seeking genericity in the selection of parameter sets: Impact on hydrological model efficiency. *Water Resources Research*, *50*(10), 8356–8366. https://doi.org/10.1002/2013WR014761

Baez-Villanueva, O. M., Zambrano-Bigiarini, M., Mendoza, P. A., McNamara, I., Beck, H. E., Thurner, J., et al. (2021). On the selection of precipitation products for the regionalisation of hydrological model parameters. *Hydrology and Earth System Sciences*, *25*(11), 5805–5837. https://doi.org/10.5194/hess-25-5805-2021

Broderick, C., Matthews, T., Wilby, R. L., Bastola, S., & Murphy, C. (2016). Transferability of hydrological models and ensemble averaging methods between contrasting climatic periods. *Water Resources Research*, *52*(10), 8343–8373. https://doi.org/10.1002/2016WR018850

Chlumsky, R., Mai, J., Craig, J. R., & Tolson, B. A. (2021). Simultaneous Calibration of Hydrologic Model Structure and Parameters Using a Blended Model. *Water Resources Research*, *57*(5), 1–22. https://doi.org/10.1029/2020WR029229

Clark, M. P., Vogel, R. M., Lamontagne, J. R., Mizukami, N., Knoben, W. J. M., Tang, G., et al. (2021). The Abuse of Popular Performance Metrics in Hydrologic Modeling. *Water Resources Research*, *57*(9), 1–16. https://doi.org/10.1029/2020WR029001

ElSaadani, M., Krajewski, W. F. F., Goska, R., & Smith, M. B. B. (2018). An Investigation of Errors in Distributed Models' Stream Discharge Prediction Due to Channel Routing. *Journal of the American Water Resources Association*, *54*(3), 742–751. https://doi.org/10.1111/1752-1688.12627

Fleischmann, A., Paiva, R., & Collischonn, W. (2019). Can regional to continental river hydrodynamic models be locally relevant? A cross-scale comparison. *Journal of Hydrology X*,

*3*, 100027. https://doi.org/10.1016/j.hydroa.2019.100027

Fleischmann, A., Paiva, R. C. D., Collischonn, W., Siqueira, V. A., Paris, A., Moreira, D. M., et al. (2020). Trade-Offs Between 1-D and 2-D Regional River Hydrodynamic Models. *Water Resources Research*, *56*(8), 1–30. https://doi.org/10.1029/2019WR026812

Fowler, K., Peel, M., Western, A., & Zhang, L. (2018). Improved Rainfall-Runoff Calibration for Drying Climate: Choice of Objective Function. *Water Resources Research*, *54*(5), 3392–3408. https://doi.org/10.1029/2017WR022466

Gharari, S., Gupta, H. V., Clark, M. P., Hrachowitz, M., Fenicia, F., Matgen, P., & Savenije, H. H. G. (2021). Understanding the Information Content in the Hierarchy of Model Development Decisions: Learning From Data. *Water Resources Research*, *57*(6). https://doi.org/10.1029/2020WR027948

Günther, D., Hanzer, F., Warscher, M., Essery, R., & Strasser, U. (2020). Including Parameter Uncertainty in an Intercomparison of Physically-Based Snow Models. *Frontiers in Earth Science*, *8*(October), 1–15. https://doi.org/10.3389/feart.2020.542599

Guse, B., Pfannerstill, M., Gafurov, A., Kiesel, J., Lehr, C., & Fohrer, N. (2017). Identifying the connective strength between model parameters and performance criteria. *Hydrology and Earth System Sciences*, *21*(11), 5663–5679. https://doi.org/10.5194/hess-21-5663-2017

Hoch, J. M., Eilander, D., Ikeuchi, H., Baart, F., & Winsemius, H. C. (2019). Evaluating the impact of model complexity on flood wave propagation and inundation extent with a hydrologic-hydrodynamic model coupling framework. *Natural Hazards and Earth System Sciences*, *19*(8), 1723–1735. https://doi.org/10.5194/nhess-19-1723-2019

Karki, R., Krienert, J. M., Hong, M., & Steward, D. R. (2021). Evaluating Baseflow Simulation in the National Water Model: A Case Study in the Northern High Plains Region, USA. *Journal of the American Water Resources Association*, *57*(2), 267–280. https://doi.org/10.1111/1752-1688.12911

Khatami, S., Peel, M. C., Peterson, T. J., & Western, A. W. (2019). Equifinality and Flux Mapping: A New Approach to Model Evaluation and Process Representation Under Uncertainty. *Water Resources Research*, *55*(11), 8922–8941. https://doi.org/10.1029/2018WR023750

Knoben, W. J. M., Freer, J. E., & Woods, R. A. (2019). Technical note: Inherent benchmark or not? Comparing Nash-Sutcliffe and Kling-Gupta efficiency scores. *Hydrology and Earth System Sciences*, *23*(10), 4323–4331. https://doi.org/10.5194/hess-23-4323-2019

Melsen, L., Teuling, A., Torfs, P., Zappa, M., Mizukami, N., Clark, M., & Uijlenhoet, R. (2016). Representation of spatial and temporal variability in large-domain hydrological models: Case study for a mesoscale pre-Alpine basin. *Hydrology and Earth System Sciences*, *20*(6), 2207–2226. https://doi.org/10.5194/hess-20-2207-2016

Melsen, L., Teuling, A. J., Torfs, P. J. J. F., Zappa, M., Mizukami, N., Mendoza, P. A., et al. (2019). Subjective modeling decisions can significantly impact the simulation of flood and drought events. *Journal of Hydrology*, *568*(November 2018), 1093–1104. https://doi.org/10.1016/j.jhydrol.2018.11.046

Mizukami, N., Rakovec, O., Newman, A. J., Clark, M. P., Wood, A. W., Gupta, H. V., & Kumar, R. (2019). On the choice of calibration metrics for "high-flow" estimation using hydrologic models. *Hydrology and Earth System Sciences*, *23*(6), 2601–2614. https://doi.org/10.5194/hess-23-2601-2019

Munier, S., & Decharme, B. (2022). River network and hydro-geomorphological parameters at 1/12 ° resolution for global hydrological and climate studies. *Earth System Science Data*, *14*(5), 2239–2258. https://doi.org/10.5194/essd-14-2239-2022

Nguyen-Quang, T., Polcher, J., Ducharne, A., Arsouze, T., Zhou, X., Schneider, A., & Fita, L. (2018). ORCHIDEE-ROUTING: Revising the river routing scheme using a high-resolution hydrological database. *Geoscientific Model Development*, *11*(12), 4965–4985. https://doi.org/10.5194/gmd-11-4965-2018

Paiva, R. C. D., Collischonn, W., & Buarque, D. C. (2013). Validation of a full hydrodynamic model for large-scale hydrologic modelling in the Amazon. *Hydrological Processes*, *27*(3), 333–346. https://doi.org/10.1002/hyp.8425

Pereira, F. F., Farinosi, F., Arias, M. E., Lee, E., Briscoe, J., & Moorcroft, P. R. (2017). Technical

note: A hydrological routing scheme for the Ecosystem Demography model (ED2+R) tested in the Tapajós River basin in the Brazilian Amazon. *Hydrology and Earth System Sciences*, *21*(9), 4629–4648. https://doi.org/10.5194/hess-21-4629-2017

Pilz, T., Francke, T., Baroni, G., & Bronstert, A. (2020). How to Tailor My Process-Based Hydrological Model? Dynamic Identifiability Analysis of Flexible Model Structures. *Water Resources Research*, *56*(8). https://doi.org/10.1029/2020WR028042

Qiao, X., Nelson, E. J., Ames, D. P., Li, Z., David, C. H., Williams, G. P., et al. (2019). A systems approach to routing global gridded runoff through local high-resolution stream networks for flood early warning systems. *Environmental Modelling and Software*, *120*(August). https://doi.org/10.1016/j.envsoft.2019.104501

Saavedra, D., Mendoza, P. A., Addor, N., Llauca, H., & Vargas, X. (2022). A multi-objective approach to select hydrological models and constrain structural uncertainties for climate impact assessments. *Hydrological Processes*, *36*(1). https://doi.org/10.1002/hyp.14446

Sepúlveda, U. M., Mendoza, P. A., Mizukami, N., & Newman, A. J. (2022). Revisiting parameter sensitivities in the variable infiltration capacity model across a hydroclimatic gradient. *Hydrology and Earth System Sciences*, *26*(13), 3419–3445. https://doi.org/10.5194/hess-26-3419-2022

Spieler, D., Mai, J., Craig, J. R., Tolson, B. A., & Schütze, N. (2020). Automatic Model Structure Identification for Conceptual Hydrologic Models. *Water Resources Research*, *56*(9). https://doi.org/10.1029/2019WR027009

Thober, S., Cuntz, M., Kelbling, M., Kumar, R., Mai, J., & Samaniego, L. (2019). The multiscale routing model mRM v1.0: simple river routing at resolutions from 1 to 50 km. *Geoscientific Model Development*, *12*(6), 2501–2521. https://doi.org/10.5194/gmd-12-2501-2019

Wang, J., Zhuo, L., Han, D., Liu, Y., & Rico-Ramirez, M. A. (2023). Hydrological model adaptability to rainfall inputs of varied quality. *Water Resources Research*. https://doi.org/10.1029/2022WR032484

Yamazaki, D., Kanae, S., Kim, H., & Oki, T. (2011). A physically based description of floodplain inundation dynamics in a global river routing model. *Water Resources Research*, *47*(4), 1–21. https://doi.org/10.1029/2010WR009726

Ye, A., Duan, Q., Zhan, C., Liu, Z., & Mao, Y. (2013). Improving kinematic wave routing scheme in Community Land Model. *Hydrology Research*, *44*(5), 886–903. https://doi.org/10.2166/nh.2012.145

Yilmaz, K. K., Gupta, H. V., & Wagener, T. (2008). A process-based diagnostic approach to model evaluation: Application to the NWS distributed hydrologic model. *Water Resources Research*, *44*(9), W09417. https://doi.org/10.1029/2007WR006716

Zhao, F., Veldkamp, T. I. E., Frieler, K., Schewe, J., Ostberg, S., Willner, S., et al. (2017). The critical role of the routing scheme in simulating peak river discharge in global hydrological models. *Environmental Research Letters*, *12*(7). https://doi.org/10.1088/1748-9326/aa7250

Zhou, L., Liu, P., Zhang, X., Cheng, L., Xia, Q., Xie, K., et al. (2023). Improving structure identifiability of hydrological processes by temporal sensitivity with a flexible modeling framework. *Journal of Hydrology*, *616*(June 2022), 128843. https://doi.org/10.1016/j.jhydrol.2022.128843

---

## Author Response (AR1)

Replies to reviews

**"To what extent does river routing matter in hydrological modelling?"**

Nicolás Cortés-Salazar, Nicolás Vásquez, Naoki Mizukami, Pablo A. Mendoza, and Ximena Vargas

We provide responses to each individual point below. For clarity, comments are given in italics, and our responses are given in plain blue text.

**Editor**

*As you have seen, the two reviewers have provided very detailed and constructive comments on your manuscript, while their overall assessment remains quite mixed. Condensing the two reviews, the single most important concern that remains in both is that the novelty and relevance that goes beyond the state-of-the art needs to be much better described and substantiated with data/results.*

*It may require some more effort to develop the manuscript to the point. I therefore strongly encourage you to substantially address all reviewer comments in detail and add the necessary detail in your analysis and discussion in major revision.*

We greatly appreciate the constructive comments provided by the Editor and the two anonymous reviewers, and the revisions in response to the comments have improved the manuscript considerably.

For this submission, we have: (i) revised our experimental setup, and now include routing parameters (in conjunction with VIC parameters) and a high-flow oriented metric in the parameter search process, following the recommendations of Reviewer #2; (ii) modified the structure of the text to provide a better explanation of the methods (in response to Reviewer #1); (iii) modified all the figures, in response to the comments from both Reviewers; (iv) refined the introduction to stress the need of this study in light of the existing literature; (v) improved the discussion to highlight the relevance of the results presented here; and (vi) improved the data availability statement, providing a Zenodo repository with all the data and codes used for this study.

It should be noted that, despite the methodological modifications, the main messages are preserved: (1) routing may alter streamflow simulations even at monthly time steps (compared to the case without routing); (2) explicitly modelling routing processes may improve calibration results and, in particular, the timing of simulated streamflow, especially for larger river basins; (3) the impact of adding the routing process goes beyond high-flow signatures (e.g., flood frequency curves), affecting hydrologic model parameters with impacts on baseflow contribution to total runoff and medium/low flow segments in the flow duration curves.

**Anonymous Referee #1**

*The authors conducted a very comprehensive sensitivity analyses of the effects of adding an additional river routing model with various schemes to a hydrological model. The publication shows promise as a great reference work for model experiment setup. In general, the publication is well written and the arguments for conducting the study are clear. The decisions made regarding the methods are well-argued (with the exception of 1) and the results are valuable for the hydrologic community. The limitation of this study are well described. It is understandable given the scope of the study and the data requirements that the authors evaluated a single catchment. For future research, I am eager to discover how the results of this study would be different in a more gentle sloping catchment or for various catchment sizes (e.g using CAMELS-CH Alvarez-Garreton et al., 2018).*

We thank the referee for meticulously reviewing our manuscript and providing several constructive suggestions. We are especially grateful for the referee's positive feedback.

*That being said, the publications needs some extra work. The main points that need attention are argumentation for hydrological model aggregation, the structure of text and figures, additional reflection on the meaning of study results, and the archiving of code and data.*

In this document, we provide our detailed responses to the reviewer's comments.

*Major comments:*

*Temporal aggregation of hydrological model results*

*In section 3.3 the authors state that for each parameter set the VIC model is run at hourly time-steps and the results are temporally aggregated to various coarser time-steps. In my opinion this is an assumption that there are no non-linear processes in time within the hydrological models. The necessity for this assumption is clear as it results in a clean model experiment. However, the authors should more clearly state this assumption and reflect on this in section 5.1 (last paragraph) and 5.2. I'm curious to read the authors response.*

This is a good point, and we fully agree with this reviewer that this limitation should be made clear. We have added the following text to make explicit the assumption that the reviewer refers to in section 4.1 (L185-189):

"To generate streamflow simulations at 2-hour, 3-hour, 4-hour, and 6-hour time steps, we aggregate hourly VIC runoff and run the mizuRoute model with all routing schemes. For example, streamflow time series at a 3-hour resolution are obtained from mizuRoute simulations using 3-hour VIC runoff time series, which are computed by temporally aggregating 1-h VIC outputs at each grid cell. It should be noted that this step requires assuming the absence of non-linear processes in time within the hydrological model."

We have added the following sentence in Section 6.1 (discussion section, L388-389):
"Accordingly, variations in the VIC model time step – which is fixed to $\Delta t = 1$ h here – may also alter the selection of parameters and performance measures (see section 6.2)"

We have also added the following text to section 6.2 (discussions section, L401-404):

"An important assumption of our experiment is the lack of non-linear processes in time within the hydrological model, in order to aggregate hourly runoff to coarser time steps. Such decision was required to isolate the impact of hydrologic model configuration from river routing decisions, and achieve a clean experimental design, though we recognize that the choice of hydrological model time step may also alter performance metrics (e.g., Bruneau et al., 1995; Wang et al., 2009)."

*Structure of text*

*The authors conducted a lot of analyses which to their credit lead to an abundance of methodology steps and results. This makes section 3.5 difficult to read and therefore it needs restructuring. I suggest to use numbering to make the steps more clear even if this disrupts the flow of the text.*
*What might also help the reader is a model run results matrix in the form of a Table that uses the same numbering. This makes it clearer for the reader what results can be expected for each type of model run configuration.*

"Figure 2 summarizes the approach used in this paper, which consists of the following steps:

a. Sample model (VIC and mizuRoute) parameter sets and obtain, for each one, streamflow times series with five routing schemes (including instantaneous runoff or no routing as the baseline) and five temporal resolutions (Figure 2a, see details in Section 4.1) at each river gage (Figure 1c). We save the parameter sets that maximize each performance metric – computed with a daily time step – at each stream gauge station for each combination of routing scheme and routing time step (Figure 1, Table 1). For the KWT scheme, we select $n$ values of 0.01 (default option), 0.03 (i.e., the spatially constant value used by Yamazaki et al., 2011) and 0.033 (which maximizes the KGE computed with daily streamflow).

b. Examine the effect of routing model configurations (i.e., routing schemes and time steps) on simulated daily hydrographs at Cautín at Cajón (Figure 2b.1), and analyze the impact of excluding the river routing process on simulated streamflow at annual, monthly, daily and sub-daily time steps (Figure 2b.2).

c. Explore the overall impacts of routing modeling decisions on performance metrics (section 4.2) computed with different temporal resolutions (Figure 2c.1). Then, we examine the sensitivity of the best metric value (achievable from the simulations with all the sampled parameter sets) to the river routing configuration across sub-basins (Figure 2c.2).

d. Characterize the effects of routing configuration on simulated annual water balance (specifically, the mean annual runoff ratio) and baseflow contribution (computed from VIC output) to total runoff (Figure 2d.1); and the selected parameter values (Figure 2d.2).

e. Analyze the effects of routing configurations on flood frequency (see details in Section 4.3, Figure 2e.1) and daily FDCs (Figure 2e.2).

The steps (a)-(d) are performed using observed and simulated data for the period April/2008-March/2012, whereas step (e) is conducted simulations for the period April/1985-March/2020. All the steps but c.1 (Figure 2) use VIC and mizuRoute parameter sets that maximize performance metrics (listed in Section 4.1) computed with simulated and observed daily flows."

Detailed descriptions are provided for parameter sampling and streamflow simulations (section 4.1, L177-193); streamflow performance metrics (section 4.2, L194-226); and (4.3) flood frequency and flow duration curves (section 4.3; L227-239). We hope that the proposed restructuring has clarified the approach used here.

*Structure of figures*

*There are issues with the presentation of the results in the figures. Overall the image quality (dpi) per figure needs to be higher. The colours used to represent the individual routing schemes are inconsistent, please check all figures.*

We have improved the image resolution and revised the colours used for the individual routing schemes.

*Figure 1: It is difficult to find the catchment on the left panel (1a). Outlining the catchment in red and using a softer tone for the country would help. The colours for elevation bands in 1b are difficult to distinguish, similar issue with the sub-basins in 1c.*

We have modified Figure 1 to include the reviewer's recommendation. We have changed the colour for Continental Chile, and highlighted the basin boundaries. Regarding panel b), the colour scale has changed and, in panel c), we have improved the contrast between streams and sub-basin delineation.

*Figure 2: Increase image quality.*

We have improved the image resolution, following the reviewer's recommendation.

*Figure 3: Highlighting the horizontal axes in red would help find the period of the zoom boxes.*

We have highlighted the horizontal axis in red for the period displayed in the zoom boxes.

*Figure 5: Colours are difficult to distinguish, suggest using the same colors for each scheme as in Figure 3. The vertical axes of each column varies, ticks for KGE are in steps of 0.2 while those of NSE are 0.4. This makes it nearly impossible to assess the relative differences in objective functions. I suggest using the same tick sizes with the exception of NSElog.*

We have harmonized the colours used for the different routing schemes in all figures.

We decided to use different axis ranges and tick sizes for each performance metric, since these are not comparable. For the particular case of NSE and KGE, the comparison between their values is not meaningful because NSE is formulated based on a reference model simulation (i.e., mean climatology), while KGE does not have one, and NSE = 0 is equivalent to KGE = -0.41 (see discussions in Knoben et al., 2019). We have incorporated these points in section 4.2 (L205-209):

"Although all these metrics range between $-\infty$ and 1, where 1 represents a perfect simulation, their values are not comparable because they differ in terms of target streamflow characteristics, and the incorporation or lack of a benchmark. For example, NSE is formulated based on a reference model simulation (i.e., mean climatology), while KGE does not have one, and NSE = 0 is equivalent to KGE = -0.41 (see discussions in Knoben et al., 2019)."

*Figure 6: There is almost no reference to the different basin areas that are shown using the horizontal axis. It would make the figure a lot clearer if only the 2770 basin area was shown and the individual schemes were plotted next to each other. I suggest placing the results for the other basin areas in the appendix.*

We have decided to keep the results for all basins in this figure to highlight that routing yields substantial improvements in the timing of simulated streamflow for larger basin areas. We now provide a more complete description on inter-basin differences in terms of river routing effects (L281-291):

"Very similar maximum KGE values are obtained with the four schemes implemented in mizuRoute, and the differences among these schemes are generally lower than 0.05 for all time steps and basins. The improvements in KGE through the inclusion of routing are explained by the enhancement of temporal correlation ($r$) and variability error (α). In particular, routing (especially Muskingum-Cunge and Diffusive Wave schemes) helps to improve $r$ values of simulated instantaneous runoff by changing the timing of high peak flows. Even more, Figure 6 shows that, in our study domain, the correlation between streamflow simulations and observations increases with basin area when routing is incorporated.
Figure 6 also shows considerable improvements in NSE across all catchments when routing is applied, especially in CatRR and CatC (i.e., the two largest river basins). Notably, differences between routing and Inst options are also obtained for NSE-log in all stream gages, demonstrating the benefits of routing beyond the simulation of high flows. %BiasFHV is very close to zero for the smallest catchment, increasing with catchment size when no routing is performed; nevertheless, in every basin, at least one routing scheme can reduce high flow biases to nearly zero."

*Figure 7: Similar to Figure 6 this figure is difficult to read. The total width of the horizontal axis does not add information, therefore I suggest to make the ticks smaller.*

We have removed the red symbols from this figure (which showed average partitioning) in order to improve the readability. We have decided to keep the same limits for x and y-axes so readers can visualize that, for KGE, NSE and NSE-log, routing has a relatively larger impact on the baseflow contribution, compared to the mean annual runoff ratio. In response to another reviewer, we have

added an additional objective function oriented to high flows (%BiasFHV), which yields parameter values that affect the mean annual water balance (L298-300):

"Precipitation partitioning (Q/P) is relatively unchanged whether routing is used or not, regardless of the routing schemes for any performance metric except %BiasFHV, for which less clear patterns in runoff ratio and flow components are obtained."

*Figure 8: Increase the image quality. I suggest to make a separate table for the objective function results.*

We have increased the image quality to improve the readability of this Figure, and decided to remove the table with metrics in each panel since that information did not contribute to the main messages of this work.

*Reflection on the meaning of study results*

*The discussion section 5.1 can be extended by reflecting more on the implications of results. For example, we understand what is happening to the hydrological model in the absence of river routing. Compensation through baseflow and no considerable change in precipitation, evapotranspiration and runoff partitioning. What is missing is, what the implication are for users and why it is important to get these parts right in hydrological model setups. This is also the case for the results in 4.4.*

To incorporate the reviewer's suggestion, we have expanded the second paragraph in section 6.1 (discussion section, L355-371):

"The results presented here show that the implementation of river routing is also relevant for medium and low flows. For example, including river routing provided higher values for NSE-log (Figures 5 and 6) – improving the simulation of low discharges – and modified the shape of the mid and low flow segments in the FDC (Figure 10), which are characteristic signatures of 'flashiness' in runoff response and long term baseflow, respectively (Yilmaz et al., 2008). The effects of river routing are also reflected in the partitioning of total runoff between baseflow and surface runoff. In fact, the results presented here show that the parameter search process compensates for the lack of routing by modifying other fluxes and state variables (Khatami et al., 2019) to increase streamflow-oriented performance metrics. For example, when the target metrics are KGE, NSE or NSE-log, excluding routing (represented by squares) forces VIC to compensate the absence of this process by delaying the runoff response with a larger contribution of baseflow to total runoff, compared to any routing schemes. In such cases, the contribution of baseflow to total runoff increases >30% when river routing is excluded, which is achieved by modifying soil parameters –including $W_s$, one of the most sensitive for baseflow in this type of hydroclimate (Sepúlveda et al., 2022) – to delay the streamflow response. This result suggests that including routing processes may impact the outcomes from drought-oriented studies, since baseflow is the primary flux sustaining streamflow during water scarcity periods (Karki et al., 2021).
Conversely, we found smaller variations in the partitioning of precipitation between evapotranspiration and runoff in the absence of river routing (Figure 7), especially for KGE, NSE and NSE-log. Hence, when models are configured to maximize these metrics to conduct hydroclimatic or water balance analyses at the annual time scale, the incorporation of routing processes is relatively less important."

*In addition, the selection of objective-function is discussed but there is no discussion on multi-objective calibration and how these might affect the results.*

In response to the reviewer's observation, we have added the following text in section 6.2 (discussion section, L400-427):

"Finally, we only considered a single-objective (e.g., NSE, KGE) parameter search based on a Monte Carlo sampling scheme. Future studies could characterize the impacts of river routing schemes exploiting single-objective optimization algorithms (e.g., Duan et al., 1992; Tolson and Shoemaker, 2007), or address multi-objective problems using Pareto principles (Yapo et al., 1998; Pokhrel et al., 2012; Shafii and Tolson, 2015). Although different behavioral parameter sets and, therefore, different internal fluxes could be obtained, we hypothesize that similar conclusions could be drawn regarding the benefits of river routing representation to achieve realistic streamflow simulations. Nevertheless, further research is needed to understand implications for catchments with different hydroclimatic regimes and physiographic characteristics."

*There is reflection needed on the relevance of the differences in objective-function values. What does a difference of xx KGE mean?*

In response to the reviewer's observation, we have added the following text in section 6.2 (discussion section, L416-420):

"In this work, we contrast the performance of different model configurations using metrics that describe specific aspects of the system's response. However, the interpretation of differences in metric values is not straightforward (Clark et al., 2021). For example, an improvement of 0.2 in KGE may be explained by a better simulation of streamflow timing and variability, at the cost of larger volume bias (e.g., see results for Cautín at Rariruca, 1305 km$^2$, Figure 6), which stresses the need to understand the information content in the metrics used for model diagnostics."

*Data*

*The authors state "The codes used in this study are available from the corresponding authors upon reasonable request". What does reasonable mean?*

*The Copernicus data policy (https://publications.copernicus.org/services/data_policy.html) states "In addition, data sets, model code, video supplements, video abstracts, International Geo Sample Numbers, and other digital assets should be linked to the article through DOIs in the assets tab."*

*In the spirit of open-science I strongly encourage the authors to do so. I leave it up to the editor to determine whether this is a requirement for publication.*

In response to this comment, we have created a repository on Zenodo (Cortés-Salazar et al., 2023 https://doi.org/10.5281/zenodo.7838673) to make our codes and data publicly available.

*Minor comments:*

*Refrain from using acronyms in figure captions. The style of figure captions is inconsistent, e.g. use of ":", or ";", or ","*

We have revised all figure captions in the manuscript, and have harmonized the writing style. We have also decided to keep the acronyms in the figure captions for consistency with the figures and the text; however, we have spelled them out to facilitate the reading.

*Lines 71-72: SWAT model is missing a reference.*

We have specified the acronym for SWAT and added a reference (Arnold et al., 1998), following the reviewer's recommendation.

*Lines 76 -80: Very long sentence, needs restructuring.*

We have re-word this sentence as follows:

"Although many past studies have shown that the choice of routing scheme affects streamflow simulations, efforts for improving their accuracy have been made by configuring hydrologic model and routing model independently. Hydrologists still focus on parameter calibration to improve discharge simulations, excluding river routing model or neglecting the potential impacts of river routing configurations (routing scheme and time step) and parameters if included (e.g., Beck et al., 2020; Newman et al., 2021). On the other hand, routing model development and evaluation uses hydrologic model output that contains varying degree of errors, becoming especially difficult in large river basins or greater spatial domains (e.g., Mizukami et al., 2016; F. Zhao et al., 2017)."

*Line 93: remove "apparently"*

We have removed this word, following the reviewer's suggestion.

*Lines 251 – 254: This is a bold claim that I would remove as it does not add value to speculate.*

We have removed this sentence, following the reviewer's recommendation.

*Line 349: "MC approach", change to machine learning approach.*

MC stands for Muskingum-Cunge. The acronym is defined in section 3.2, but here we decide to spell out to avoid confusion among readers. Thanks for this observation!

*Personal dislike of the use of the word "indeed" throughout the publication.*

We have removed the word 'indeed' from the manuscript, following the reviewer's recommendation.

**Anonymous Referee #2**

*I have finished my review of the paper "To what extent does river routing matter in hydrological modeling", by Cortés-Salazar et al., submitted to HESS. This generally well-written paper attempts to examine the influence of routing algorithm and time step on model performance using subsets of 3500 different runoff regimes generated using the VIC hydrological model. Very mild differences were found between algorithm and time step choices, with one exception: where routing was not simulated, the performance was consistently poor relative to models which simulate routing. Unfortunately, this is not a very compelling result.*

*While the approach was rigorous in the sense that it compiled data from thousands of model simulations, it suffers from a number of critical methodological issues.*

We thank the referee for his/her time on reviewing our manuscript and providing several constructive suggestions. We provide our detailed responses to all the reviewer's comments.

*I discuss a few of these major issues below:*

*Routing is most influential on peak magnitude and timing of large events; both of these are poorly captured by integrated hydrograph metrics such as KGE and NSE. Peak flow differences after calibrating the routing models would be a much more useful metric for evaluating routing model performance. By using integrated measures such as KGE, the critical differences between routing algorithms are not discernable (as seen in nearly all of the reported results).*

We agree with this reviewer that including additional high-flow related metrics would increase the impact of this work. Therefore, we have added the bias in the high flow segment of the flow duration

curve (Yilmaz et al., 2008) as an additional calibration metric. The justification for including this metric and related equation are now included in section 4.2 ("Performance metrics", L215-218) of the revised manuscript.

In this work, we decide to include KGE, NSE and log-NSE in the analysis, since these integrated metrics are not only of interest for the hydrologic modeling community – especially for parameter calibration and evaluation (e.g., Fowler et al., 2018; Knoben et al., 2019; Clark et al., 2021) –, but also for the river routing community. Indeed, several examples of river routing scheme assessments can be found using the KGE (e.g., Pereira et al., 2017; Hoch et al., 2019; Qiao et al., 2019; Thober et al., 2019; Munier and Decharme, 2022), NSE (e.g., Yamazaki et al., 2011; Ye et al., 2013; Zhao et al., 2017; ElSaadani et al., 2018; Nguyen-Quang et al., 2018; Fleischmann et al., 2019, 2020) and even the NSE using flows in logarithmic space (Paiva et al., 2013). We believe that the joint analysis of traditional streamflow performance metrics and high flow metrics helps to expand the target audience of this work. We have provided a proper justification for selecting these metrics in the new section 4 ("Experimental setup", L209-214) of the revised manuscript.

*Each of the figures in the report are reporting ALL of the output from the simulations, regardless of whether it is important or interpretable or worthy of interpretation.*

Figure 5 is the only one that originally contained results from all the parameter samples, though we have modified this to address the following reviewer's comment (see next response).

*For instance, figure 5 reports KGE, NSE, and NSE of log transformed flows for all 3500 simulations with multiple timesteps, multiple routing schemes. In addition to the only interpretable result from this figure is that no routing is outperformed by routing, there is little utility in comparing NSE values of 0.2-0.3 (the approximate median of these simulations) -differences in NSE below about 0.5 are nearly arbitrary in that a hydrograph with an NSE of 0.2 may not be visibly preferable to an NSE of 0.05. The only feature of this plot referred to in the text was the maximum metric value. Why not simply report that?*

The original motivation of Figure 5 was to illustrate the impact that incorporating river routing modelling may have on streamflow performance metrics across the VIC parameter space. Nevertheless, we fully agree that it would make more sense to simply illustrate the results for a subset of behavioural parameter sets (as we did in Figure 7). Hence, in the revised manuscript we select and report only the best 1% of runs, following the approach of Melsen et al. (2016).

*Critically, because the parameters of the VIC model are arbitrary, the comparisons of even the best models are in effect the results of Monte Carlo calibration, the least efficient optimization approach.*

We decide to use a Monte Carlo parameter sampling approach because, rather than seeking for an optimal parameter set, our primary goal is to assess the impacts of different river routing configurations on streamflow metrics across the model parameter space. In particular, Latin Hypercube Sampling is a common strategy to sample the parameter space and identify behavioural parameter sets for a specific target metric (e.g., Andréassian et al., 2014; Broderick et al., 2016; Melsen et al., 2016, 2019; Guse et al., 2017; Khatami et al., 2019). We have clarified this in section 4.1 ("Parameter sampling and streamflow simulations") of the revised manuscript, adding the following text (L178-181):
"Since we aim to examine the impacts of different routing schemes on streamflow performance metrics across the parameter space, rather than seeking an optimal parameter set, we use the Latin Hypercube Sampling (LHS) method, which is a common strategy to sample the parameter space and identify behavioral sets for specific target metrics (e.g., Andréassian et al., 2014; Broderick et al., 2016; Melsen et al., 2016, 2019; Guse et al., 2017; Khatami et al., 2019)."

*Comparing the 'best' models when these are not rigorously determined to be the actual best for each algorithm (rather than a sampling error) is problematic. For this comparison to be rigorous, I don't see how to do this without simultaneous calibration of routing and land surface parameters, an issue the authors acknowledge in section 5.2.*

River routing parameters were excluded from the original setup in order to make a clean numerical experiment (recall that the baseline model has no routing module). However, we appreciate the reviewer's concern, and in response to this comment we have performed new LHS experiments that include river routing parameters (see response to the following comment).

*In practice, the routing parameters (such as Manning's n) would be calibrated in conjunction with VIC model parameters, likely further diminishing any incremental performance differences between the routing models.*

In order to address this point, now we sample 3500 model parameter sets considering the 13 VIC parameters identified by Sepúlveda et al. (2022) as the most sensitive, and routing model parameters: one (the Manning roughness coefficient) for the KWT, MC and DW methods, and two for the IRF method (Table 2). The new Figure 5 (attached) now contains the 1% best parameter sets.

Table 2. Model parameters sampled in this study.

| Parameter | Units | Lower value | Upper value | Description |
|---|---|---|---|---|
| Infilt | - | 0.01 | 0.99 | Variable infiltration curve parameter |
| $D_s$ | - | 0.1 | 0.9 | Fraction of $D_{s\,max}$ where non-linear baseflow occurs |
| $D_{s\,max}$ | mm/d | 0.1 | 300 | Maximum velocity of baseflow |
| $W_s$ | - | 0.1 | 0.9 | Fraction of maximum soil moisture where non-linear baseflow occurs |
| expt | - | 3.1 | 10 | Exponent of Campbell's equation for hydraulic conductivity |
| $d_{max}$ $d_1$ $d_2$ $d_3$ | m | 0.5 $0.05\,d_{max}$ $0.21\,d_{max}$ $0.74\,d_{max}$ | 5 $0.2\,d_{max}$ $0.7\,d_{max}$ $0.1\,d_{max}$ | Depth of soil layers 1, 2 and 3 |
| $K_{sat}$ | mm/d | 1 | 1000 | Saturated hydraulic conductivity |
| $T_{max,snow}$ | (°C) | -10 | 10 | Maximum temperature for snowfall |
| $\alpha_{thaw}$ | - | 0.75 | 0.90 | Decay of albedo |
| $\alpha_{new}$ | - | 0.85 | 0.95 | Maximum albedo for fresh snow |
| $n$ | $s/m^{1/3}$ | 0.024 | 0.075 | Roughness coefficient of Manning (Barnes, 1967) |
| $C$ | $m/s$ | 0.25 | 10 | Advection coefficient (Allen et al., 2018) |
| $D$ | $m^2/s$ | 200 | 4000 | Diffusion coefficient (Melsen et al., 2016) |

Overall, the results show a clear difference between including routing and no routing (Inst), which becomes more evident for performance metrics computed at smaller temporal resolutions. Moreover, none of the 1% best parameter sets for KGE and NSE with instantaneous runoff could produce better performance than including routing. On the other hand, the choice of routing time step is comparatively less influential for a given metric time step. The maximum KGE spans 0.69-0.73 for instantaneous runoff, increasing to 0.8 or more when routing is included. Similar improvements are observed for NSE, with increments that can be larger than 0.3 NSE values (e.g., 1 h). Compared to the former metric, routing yields smaller benefits for NSE-log and less noticeable differences among

routing configurations, mainly due to the minor influence of high flow values on the metric. In all cases, a larger spread in high-flow biases is obtained with instantaneous runoff (compared to routing schemes), indicating that many VIC parameter sets do not compensate for the lack of river routing to obtain accurate high flow simulation. Finally, the results show that the impact of representing river routing on %BiasFHV is more relevant when this metric is computed at hourly resolution, approaching to zero as the metric time step increases. All these points are made in L265-276.

[Figure]

Figure 5. Impact of routing scheme and routing time step on performance metrics (rows) for the period April/2008-March/2012 at Cautín at Cajón, computed with different discharge temporal resolutions (columns) and the 1% best parameter sets among those obtained through Latin Hypercube Sampling (see text for details). The results are presented for instantaneous runoff (Inst), Impulse Response Function (IRF), Kinematic Wave Tracking (KWT), Muskingum-Cunge (MC) and Diffusive Wave (DW)

Since the experimental setup has been modified, we have updated all the figures as well as the text accordingly. In any case, the main messages are preserved, stressing that the potential impacts of routing configurations go beyond high-flow metrics, affecting simulated streamflow even at monthly time scales, along with medium and low flows.

*The fundamental results discussed here are obvious without the testing herein.*

We provide individual responses to all the reviewer's comments on this matter below:

*Routing is better than no routing.*

Of course, the incorporation of as many hydrological processes as observed for the natural system of interest – including river routing – is desirable to achieve "fidelius" (Gharari et al., 2021) simulations

in hydrology and land surface models. Nevertheless, the degree of improvement for application-specific metric is not obvious, considering that the interplay between model structural uncertainty (here, provided by different routing schemes) and parametric uncertainty is not fully understood and, therefore, it is a topic of active research (e.g., Günther et al., 2020; e.g., Pilz et al., 2020; Spieler et al., 2020; Chlumsky et al., 2021; Zhou et al., 2023). In particular, it is not clear to which degree the perturbation of hydrological model parameters can compensate for the lack of river routing representation – a gap that this work intends to reduce and is now highlighted in the introduction (L78-87):

"Although many past studies have shown that the choice of routing scheme affects streamflow simulations, efforts for improving their accuracy have been made by configuring hydrologic model and routing model independently. Hydrologists still focus on parameter calibration to improve discharge simulations, excluding river routing model or neglecting the potential impacts of river routing configurations (routing scheme and time step) and parameters if included (e.g., Beck et al., 2020; Newman et al., 2021). On the other hand, routing model development and evaluation uses hydrologic model output that contains varying degree of errors, becoming especially difficult in large river basins or greater spatial domains (e.g., Mizukami et al., 2016; F. Zhao et al., 2017). Further, the key role of river routing parameters to reproduce observed streamflow characteristics has been previously recognized (Boyle et al., 2001; Butts et al., 2004; Sheikholeslami et al., 2021), highlighting the need for joint (i.e., hydrological and routing) parameter search strategies to characterize the benefits of routing configurations and potential compensatory effects in reproducing application-specific metrics."

*Low flows are not as impacted by routing differences.*

In principle, the effects of a specific model configuration (i.e., routing scheme or routing time step) on certain processes (e.g., high or low flows) are not comparable, since these are assessed with different performance metrics (e.g., Mizukami et al., 2019; Zhou et al., 2023). Additionally, the results presented here show that the implementation and configuration of river routing schemes are also relevant for medium and low flows. For example, including river routing provided higher values for NSE-log (Figures 5 and 6) – improving the simulation of low discharges – and modified the shape of the mid and low flow segments in the FDC (Figure 10), which are characteristic signatures of 'flashiness' in runoff response and long term baseflow, respectively (Yilmaz et al., 2008). The effects of river routing are also reflected in the partitioning of total runoff between baseflow and surface runoff. In fact, the results presented here show that the parameter search process compensates for the lack of routing by modifying other fluxes and state variables (Khatami et al., 2019) to increase streamflow-oriented performance metrics. For example, when the target metrics are KGE, NSE or NSE-log, excluding routing (represented by squares) forces VIC to compensate the absence of this process by delaying the runoff response with a larger contribution of baseflow to total runoff, compared to any routing schemes. In such cases, the contribution of baseflow to total runoff increases >30% when river routing is excluded, which is achieved by modifying soil parameters –including $W_s$, one of the most sensitive for baseflow in this type of hydroclimate (Sepúlveda et al., 2022) – to delay the streamflow response. This result suggests that including routing processes may impact the outcomes from drought-oriented studies, since baseflow is the primary flux sustaining streamflow during water scarcity periods (Karki et al., 2021). All these points are now highlighted in the discussion (Section 6.1, L355-367).

*Parameter compensation occurs in hydrological models.*

We agree that this is a well-known issue in the hydrologic modelling community, including compensation of hydrologic model parameters on forcing errors (e.g., Baez-Villanueva et al., 2021; Wang et al., 2023) and hydrologic model structure deficiencies (e.g., Saavedra et al., 2022). To the best of our knowledge, however, this is the first study that characterizes how hydrologic model parameter estimation can compensate for the absence of river routing representation, with focus on streamflow performance metrics, parameter values, flux partitioning and signatures used for water

resources applications (in our case, flood frequency and flow duration curves), contributions that are highlighted in the introduction (L91-98). Our results show that parameter compensation is not trivial, and depends on the calibration metric and fluxes analysed (Figure 7).

*Routing impacts are not visible when averaged over a monthly time step.*

The results presented here demonstrate exactly the opposite. Indeed, the second row in Figure 4 shows that the difference in monthly flows between routed vs. non-routed flows can be as large as 27 $m^3$/s. We thank the reviewer for this comment, and we have modified the text in order to clarify this point (L257-259):

"At monthly time steps, the differences between routed and instantaneous runoff reduce considerably, although these still can be as large as 27 $m^3$/s (i.e., a 10% difference using routed runoff as the reference)."

*Models with instantaneous routing (i.e., no routing) have higher flows.*

We agree (and also shown in Figures 9 and 10) that excluding routing yields higher flows. However, errors observed in the simulations from the VIC model without routing are actually timing errors (due to the absence of travel time in the channel) and variability error (due to no attenuation accounted for). Importantly, the new Figure 6 shows that, when the routing process is explicitly modelled, improvements in streamflow timing are more evident as catchment area increases. The above errors are often assumed to become negligible when temporally aggregating non-routed flows. Nevertheless, this study unveils that the impacts of river routing go beyond the simulation of high flows at sub-daily or daily time steps, affecting streamflow simulations even at the monthly time scale (see previous response). The extent to which these effects propagate across modelling decisions is not trivial and has not been previously documented.

*As is, I do not see an additional contribution from this work beyond that which exists in the literature. It is for the above reasons that I must recommend rejection.*

We regret that the reviewer failed to agree with us regarding the contribution of the manuscript. We have addressed all the critiques raised in this review, and have conveyed more clearly the relevance of our work in the revised version.

---

## Author Response (AR2)

Replies to reviews

**"To what extent does river routing matter in hydrological modelling?"**

Nicolás Cortés-Salazar, Nicolás Vásquez, Naoki Mizukami, Pablo A. Mendoza, and Ximena Vargas

We provide responses to each individual point below. For clarity, comments are given in italics, and our responses are given in plain blue text.

**Editor**

*Thank you very much for your thoughtful revisions. Based on the assessment of one reviewer and my own reading of your manuscript, I believe that your work has strongly benefited from your additional efforts. However, as the reviewer points out, the manuscript could be further strengthened by making it more concise and developing a clearer punchline: what can and does your analysis really show? Reducing the material that does not really support your argument will help the reader to more strongly appreciate the findings of your experiment. That is not to say that all other material needs to be omitted, but it can be made available for the interested reader as supplementary material.*
*I think these adaptations can be made in an additional round of minor revisions..*

Thank you very much for your positive feedback. In response to your suggestion, we have created a Supporting Information file that contains the full results, and have made the following modifications in the manuscript:
- Figure 5: we now show results for only two metric time steps (1 h and 24 h).
- Figures 7, 9 and 10: we now show results for only three routing time steps (1 h, 3 h and 6 h).

Accordingly, we have made slight modifications to the text (L265-266, L295-296 and L345). I should be noted that none of these changes affect the main messages of this work.

Thank you again for your time and consideration. We look forward to hearing from you.

Pablo A. Mendoza and co-authors.